# HCK induces macrophage activation to promote renal inflammation and fibrosis via suppression of autophagy

Man Chen[1,2,3], Madhav C. Menon[4], Wenlin Wang[1], Jia Fu [1], Zhengzi Yi[1], Zeguo Sun[1], Jessica Liu[1], Zhengzhe Li[1], Lingyun Mou [1], Khadija Banu [4], Sui-Wan Lee[5], Ying Dai[5], Nanditha Anandakrishnan [1], Evren U. Azeloglu [1], Kyung Lee [1], Weijia Zhang[1], Bhaskar Das[6] ✉, John Cijiang He [1,7] ✉ & Chengguo Wei [1] ✉

Renal inflammation and fibrosis are the common pathways leading to progressive chronic kidney disease (CKD). We previously identified hematopoietic cell kinase (HCK) as upregulated in human chronic allograft injury promoting kidney fibrosis; however, the cellular source and molecular mechanisms are unclear. Here, using immunostaining and single cell sequencing data, we show that HCK expression is highly enriched in pro-inflammatory macrophages in diseased kidneys. HCK-knockout (KO) or HCK-inhibitor decreases macrophage M1-like pro-inflammatory polarization, proliferation, and migration in RAW264.7 cells and bone marrow-derived macrophages (BMDM). We identify an interaction between HCK and ATG2A and CBL, two autophagy-related proteins, inhibiting autophagy flux in macrophages. In vivo, both global or myeloid cell specific HCK-KO attenuates renal inflammation and fibrosis with reduces macrophage numbers, pro-inflammatory polarization and migration into unilateral ureteral obstruction (UUO) kidneys and unilateral ischemia reperfusion injury (IRI) models. Finally, we developed a selective boron containing HCK inhibitor which can reduce macrophage pro-inflammatory activity, proliferation, and migration in vitro, and attenuate kidney fibrosis in the UUO mice. The current study elucidates mechanisms downstream of HCK regulating macrophage activation and polarization via autophagy in CKD and identifies that selective HCK inhibitors could be potentially developed as a new therapy for renal fibrosis.

According to the Centers for Disease Control and Prevention, more than 10 percent of American adults have chronic kidney disease (CKD). With increases in comorbidities, diabetes, obesity, atherosclerosis and hypertension, the estimated number of cases with CKD will continue to grow[1,2]. Chronic renal allograft injury (CAI), a major cause of CKD in transplant patients, is characterized by interstitial fibrosis/tubular atrophy (IF/TA) and is the most common cause of long-term renal allograft failure in the first post-transplant decade. Kidney inflammation and fibrosis are the common pathways mediating the progression of CKD and CAI[3]. However, the underlying mechanisms remain obscure, and the treatment is lacking. It has been shown that excessive and prolonged inflammation is a major driver of fibrosis[4,5]. Macrophage is known to play a major role in regulation of renal inflammation and fibrosis by producing pro-inflammatory cytokines and pro-fibrosis

factors[6]. Although multiple profibrotic factors have been identified, these experimental interventions have not been translated into clinical practice. Recent clinical trial using anti-TGFβ monoclonal antibody has failed in patients with diabetic kidney disease. TGF-β also has anti-inflammatory properties[7–9] and a complete blockade of TGF-β may enhance inflammatory response[10]. Therefore, there is an urgent need to develop better drug targets for treatment of renal inflammation and fibrosis in CKD patients.

SRC family kinases (SFKs) belong to a family of non-receptor protein tyrosine kinases and have been implicated in the regulation of numerous cellular signaling and biological effects[11,12]. Previous studies have demonstrated the important role of SFKs in the development of various fibrosis-related chronic diseases such as idiopathic pulmonary fibrosis, liver fibrosis, renal fibrosis, and systemic sclerosis[13]. Our recent unbiased screening studies revealed that hematopoietic cell kinase (HCK), a SFK member, promotes renal fibrosis pathway and dasatinib, a non-selective inhibitor of HCK attenuates renal fibrosis in mice with unilateral ureteral obstruction and lupus nephritis[14]. However, how HCK contributes to renal fibrosis remains unclear. Several studies suggest that HCK plays a major role in regulating macrophage functions of polarization, migration, and proliferation[11,15,16]. Since single cell RNA sequencing (scRNA-seq) analysis suggests that HCK expresses mostly in pro-inflammatory infiltrating macrophages in the kidney, we studied here the role and the mechanisms of HCK in regulation of macrophage function in the context of renal fibrosis. In RAW264.7 and BMDM cells, we find that inhibition of HCK activity by KO or inhibitor decreases macrophage pro-inflammatory polarization, proliferation, and cell migration by interacting with ATG2A and CBL, two autophagy-related proteins. In vivo studies confirm the data that HCK-KO attenuates renal inflammation and fibrosis with reduces pro-inflammatory polarization and infiltration in mouse kidney models. In addition, we developed a more selective boron containing HCK inhibitor and we show that this inhibitor has anti-fibrosis and anti-inflammatory effects in cells and animal models of kidney disease.

## Results

### HCK expression was associated with inflammation and fibrosis in CAI and CKD and specifically expressed in macrophages

Our previous study showed that HCK was significantly upregulated in renal allografts of patients with CAI and associated with chronic allograft damage index (CADI) and interstitial fibrosis and tubular atrophy (IF/TA) scores, while high CADI-scores at 12-months associated with graft loss[17]. Here, we further demonstrated that HCK expression was positively correlated with renal inflammation and/or fibrosis as represented by scoring of i + t (i, interstitial inflammation; t, tubulitis) and ci+ct (ci, interstitial fibrosis, ct, tubular atrophy) with HCK expression at 12 months post transplantation (Fig. 1A). Immunohistochemistry staining showed that HCK and phospho-HCK (Y410) positively stained cells were co-localized with macrophage marker CD68 in the interstitial area suggesting that HCK was expressed mostly in macrophages. Quantitative analysis indicated that both activity of HCK (phosphorylation of HCK at Y410) and protein level of HCK were upregulated in macrophages in the biopsies from patients with CAI (Fig. 1B, C). Consistently, analysis of scRNA-seq datasets showed that HCK is predominantly expressed in macrophages and the expression level was upregulated from health to diseased kidneys (Supplementary Fig. S1A–G). In addition, HCK was mainly expressed in inflammatory (CCR2), infiltrating (Ly6) and proliferating (Mki67) macrophages subpopulation. (Supplementary Fig. S1H). These data suggest that HCK expression is enriched in activated pro-inflammatory macrophages in kidney disease conditions.

### HCK regulated macrophage polarization, proliferation, and migration

Based on these data we investigated the role of HCK in regulating macrophage function during kidney injury. Cytokines are essential to the functions of macrophages. We performed a cytokine array using bone marrow-derived macrophages (BMDM) isolated from WT and HCK KO mice to assay effects of HCK. The inflammatory cytokines including CD54, IL-6, CXCL10, CXCL1, CCL2, CCL12, CXCL9, TIMP1 and TNF-α were decreased in HCK-KO and dasatinib (HCK inhibitor) treated BMDM, compared to WT BMDM (Fig. 2A, B). qPCR demonstrated that mRNA levels of M1 inflammatory markers (TNF-a, iNOS, IL1b and CCL2) were decreased, while M2 makers (Arg1, Mrc1, Fizz-1 and Ym1) were increased in HCK KO BMDM comparing to WT BMDM after stimulation with LPS/INFγ or IL4 (Fig. 2C). We also utilized WB and found HCK KO decreased macrophage polarization to M1 and increased polarization to M2 phenotype with markers iNOS and CD206 (Fig. 2D).

Next, we tested the effect of HCK on macrophage proliferation and migration. We found HCK KO and treatment with dasatinib both decreased BMDM proliferation with MTT (Fig. 3A). We also performed flow cytometry with propidium iodide (PI) in BMDM and found HCK KO increased G0/G1 phases while decreased G2/M phases (Fig. 3B), indicating that inhibition of HCK activation decreased BMDM cell proliferation. Click-iT™ EdU cell proliferation assay and Ki67 IF staining (Fig. 3C) for cell proliferating were also performed. EdU and Ki67 positive BMDMs were decreased in HCK KO group, compared to WT control cells. We performed scratch assay in BMDM cells and found that HCK KO significantly decreased migration of these cells significantly (Fig. 3D). We also utilized 3D transwell assay to measure the cell invasion with WT and HCK KO BMDM. Invasion of BMDMs through Matrigel® Matrix coated transwell was decreased with HCK KO (Fig. 3E). Moreover, long-time-live-cell imaging showed that 3D migration velocity in Matrigel for HCK KO (1.588 μm/minute ± 0.7382 SD) and dasatinib treatment (1.838 μm/minute ± 1.045 SD) were decreased, compared to WT control in BMDM (2.469 μm/minute ± 1.700 SD) (Fig. 3F, Supplementary Movies 1–12). HCK's effects on macrophage migration were partially through integrin signaling pathway as we demonstrated that HCK KO in BMDMs decreased their spreading size by F-actin staining and phosphorylation of SYK, a key regulator of integrin signaling, by western blot analysis (Supplementary Fig. S2). Together, these data indicated that HCK plays a key role in regulating macrophage activity. Inhibiting HCK activity by KO or inhibitor could reduce the macrophage proliferation, migration, and polarization to M1-like proinflammatory state.

### HCK regulates macrophage activity through autophagy by interacting with ATG2A and CBL

To explore the mechanisms of how HCK regulates macrophage function, we screened HCK interacted proteins by performing immunoprecipitation (IP) and mass spectrometry (MS) with anti-V5-tag mAb-magnetic beads with overexpression of HCK-V5, FYN-V5 and SRC-V5. We identified 339, 403 and 715 proteins interacting with HCK, FYN, and SRC, respectively (Table 1). Interestingly, we identified ATG2A and CBL as the top-ranked proteins to interact with HCK but not with SRC and FYN. ATG2A is a critical protein for autophagosome formation[18] and lipid droplets phagocytosis[19]. CBL can regulate autophagy by suppressing mTOR and enhancing ERK1/2 activation, which are master regulators of autophagy activity[20]. In RAW264.7 macrophage cells, we confirmed that endogenous HCK interacted with both ATG2A and CBL, as shown by IP combined with immunoblotting (Fig. 4A). Since autophagy plays important roles in macrophages for cytokine release, polarization, phagocytosis[21], we hypothesized that HCK regulated macrophage activity through inhibition of autophagy via its interacting autophagy related

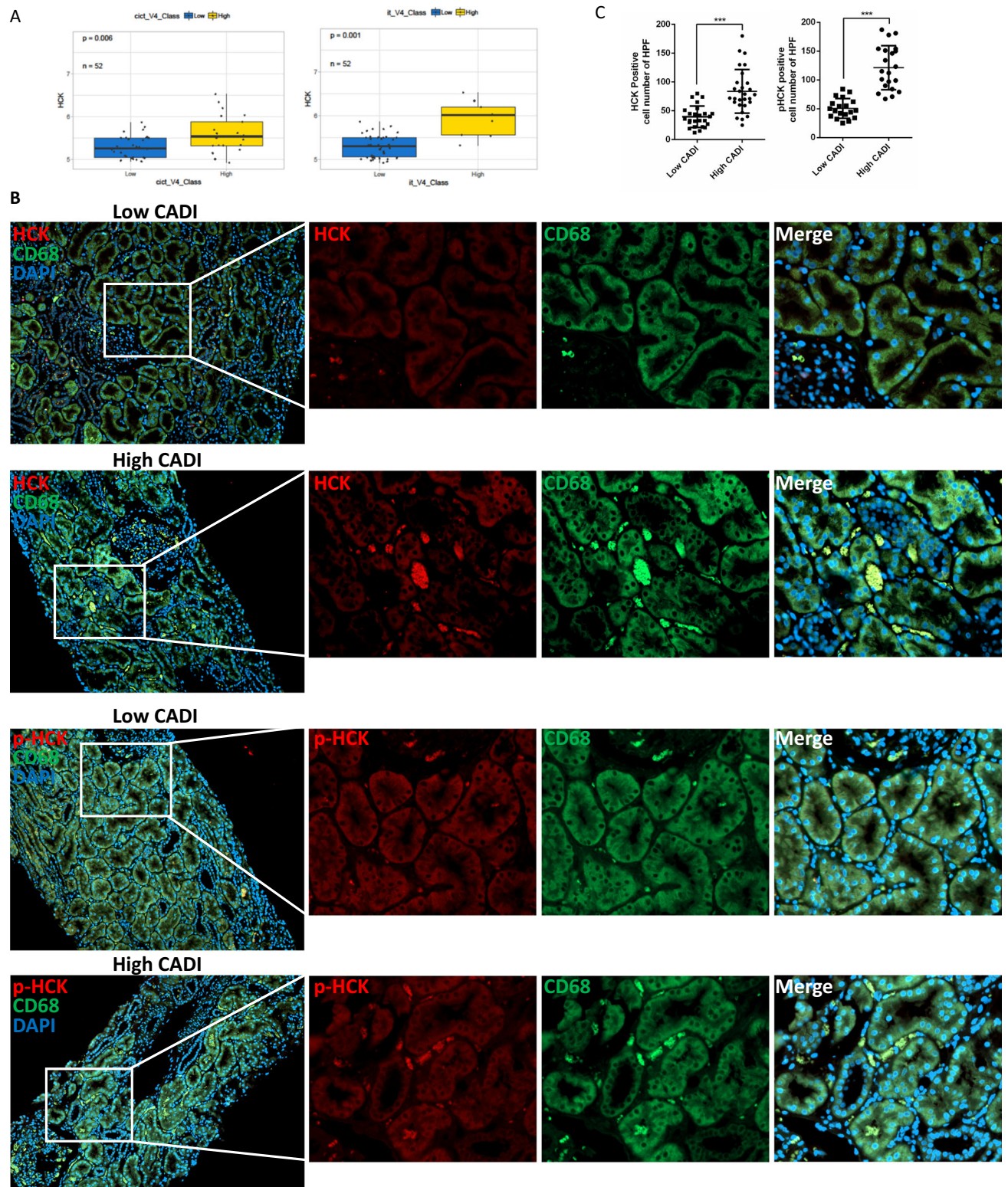

**Fig. 1 | HCK's expression was associated with inflammation and fibrosis in allograft biopsies. A** The expression of HCK was up-regulated in high i + t and ci+ct scores (>=2 vs. <2) vs low scores in allograft biopsies at 12 months after transplantation. **B** IHC Staining of phosphorylated-HCK, HCK and CD68 in high chronic allograft damage index (CADI) and low CADI allograft kidneys. **C** Quantification ($n = 4$ biopsies/group; 5-6 random fields/biopsy) of the phosphorylated-HCK and HCK positive cells in the high and low CADI biopsies. i, interstitial inflammation; t, tubulitis. ci, interstitial fibrosis; ct, tubular atrophy. ***$p < 0.001$. Source data are provided as a Source Data file.

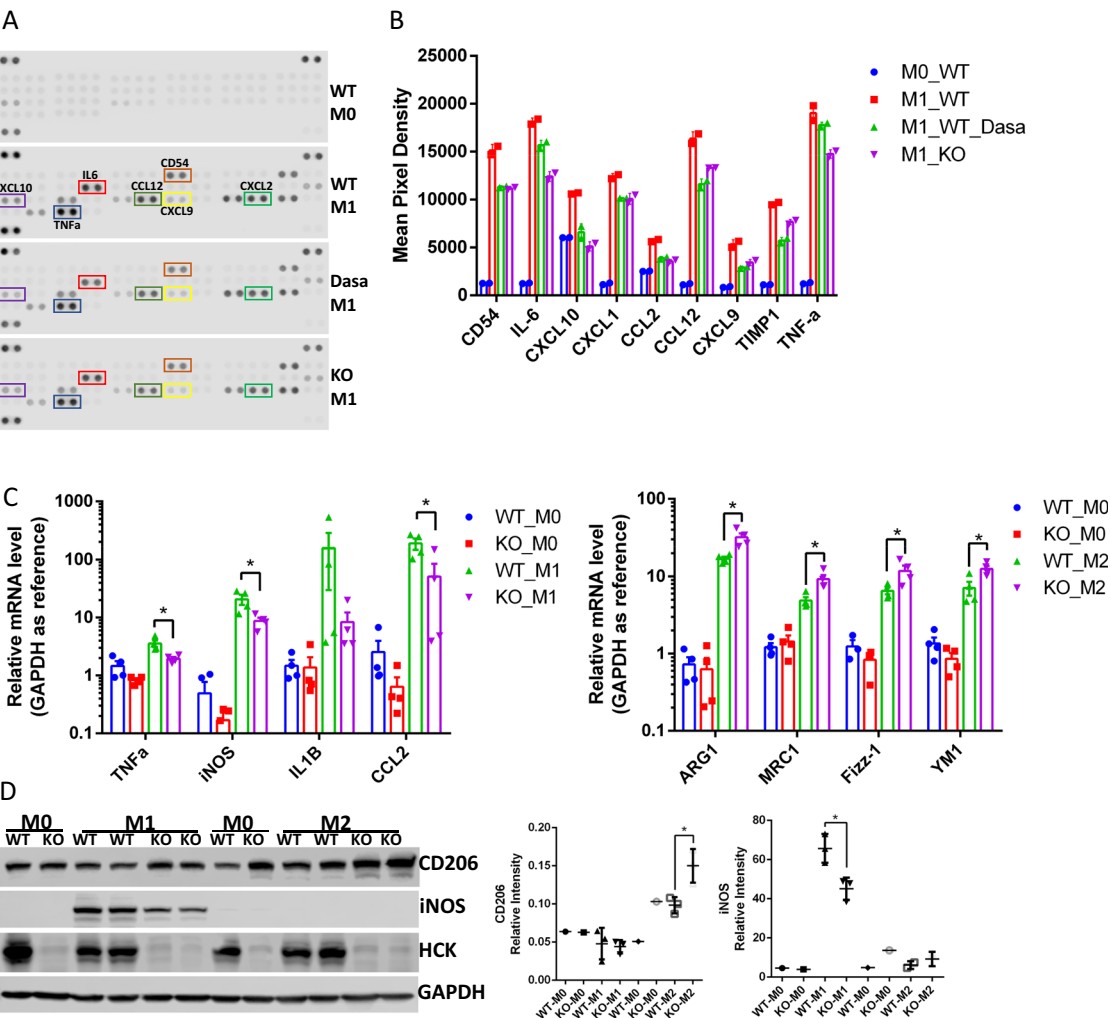

**Fig. 2 | Inhibiting HCK activity by inhibitor and HCK KO regulated macrophage polarization in BMDM and Raw264.7. A** Cytokine antibody array was performed with cultured medium from control, dasatinib treated and HCK KO BMDMs. **B** Quantification of cytokines with the average of two dots pixel density from cytokine assay. qPCR (**C**) and western blot (**D**) demonstrated that HCK KO

decreased M1 and increased M2 macrophage polarization in BMDM cells. BMDM were isolated from WT and HCK KO mice and induced with LPS/IFNγ or IL-4 after culture for 7 days. **C, D** $*p < 0.05$. Dasa: dasatinib. Source data are provided as a Source Data file.

proteins. We first measured the autophagy activity by utilizing LC3 HiBiT reporter assay from Promega with HCK overexpression and knockdown. Overexpression of HCK increased total LC3 levels indicating that HCK inhibited autophagy flux[22,23], while knockdown of HCK depleted LC3 level indicating a restoration of autophagy flux (Fig. 4B). In addition, we determined the autophagy activity in BMDM isolated from HCK KO and WT mice. We found that KO of HCK increased autophagy activity as reflected by increased LC3II/ LC3I ratio and decreased SQSTM1/P62 in basal condition and in autophagy inhibiting condition with 3MA treatment (Fig. 4C). We found phospho-PI3K and phospho-AKT decreased in HCK KO BMDMs. PI3K/AKT pathway is a known regulator of autophagy pathway through mTORC1. However, there were no significant differences for phospho-Erk1/2 and phospho-EGFR with HCK KO (Fig. 4D, E). We also demonstrated that the effect of HCK KO on macrophage polarization was abrogated by treating cells with PP242, an autophagy inducer (Fig. 4F). Together with Fig. 2D which shows that HCK KO decreasing inflammatory activation (M1) and increasing alternative activation (M2), these data suggest that HCK's effects on macrophage polarization are mainly regulated through autophagy. All these data indicated that HCK regulates macrophage activity through inhibition of autophagy pathway.

## Knockout of HCK in macrophages attenuates renal fibrosis in the UUO mouse model

To study the role of HCK in vivo in mouse kidney disease, we developed HCK exon3 loxp flanked transgenic mouse at EuMMCR in Germany (HCK ES Cell Clone: HEPD0510, Fig. 5A). These mice were crossed with CMV-Cre and LysM-Cre mice to generate global and myeloid cell specific HCK KO mice. KO of HCK was confirmed by western blot in BMDM isolated from CMV-Cre-HCK-KO and LysM-Cre-HCK-KO mice (Fig. 5B). Next, we performed UUO model in global (CMV-Cre-HCK-KO) and myeloid cell specific (LYZ2-Cre-HCK-KO) homozygous KO mice and WT mice. Here, both global and myeloid cell specific HCK KO mice attenuated tubulointerstitial fibrosis, as shown by decreased mRNA levels of profibrotic markers (Fig. 5C) and Masson's trichrome staining (Fig. 5D, E). Total macrophage cell number as stained by F4/80 decreased in UUO kidneys of both global and macrophage specific HCK KO mice, compared with WT mice (Fig. 5F). Western blot demonstrated that both M1 (iNOS) and M2 (CD206) markers decreased in HCK KO UUO kidneys due to the reduction of total macrophage number (Fig. 5G). When adjusted for macrophage numbers (ie F4/80-positive cell numbers), M1 marker iNOS was decreased and M2 marker CD206 increased in the KO mice indicating that HCK KO could regulate macrophage polarization by skew from M1 inflammatory phenotype

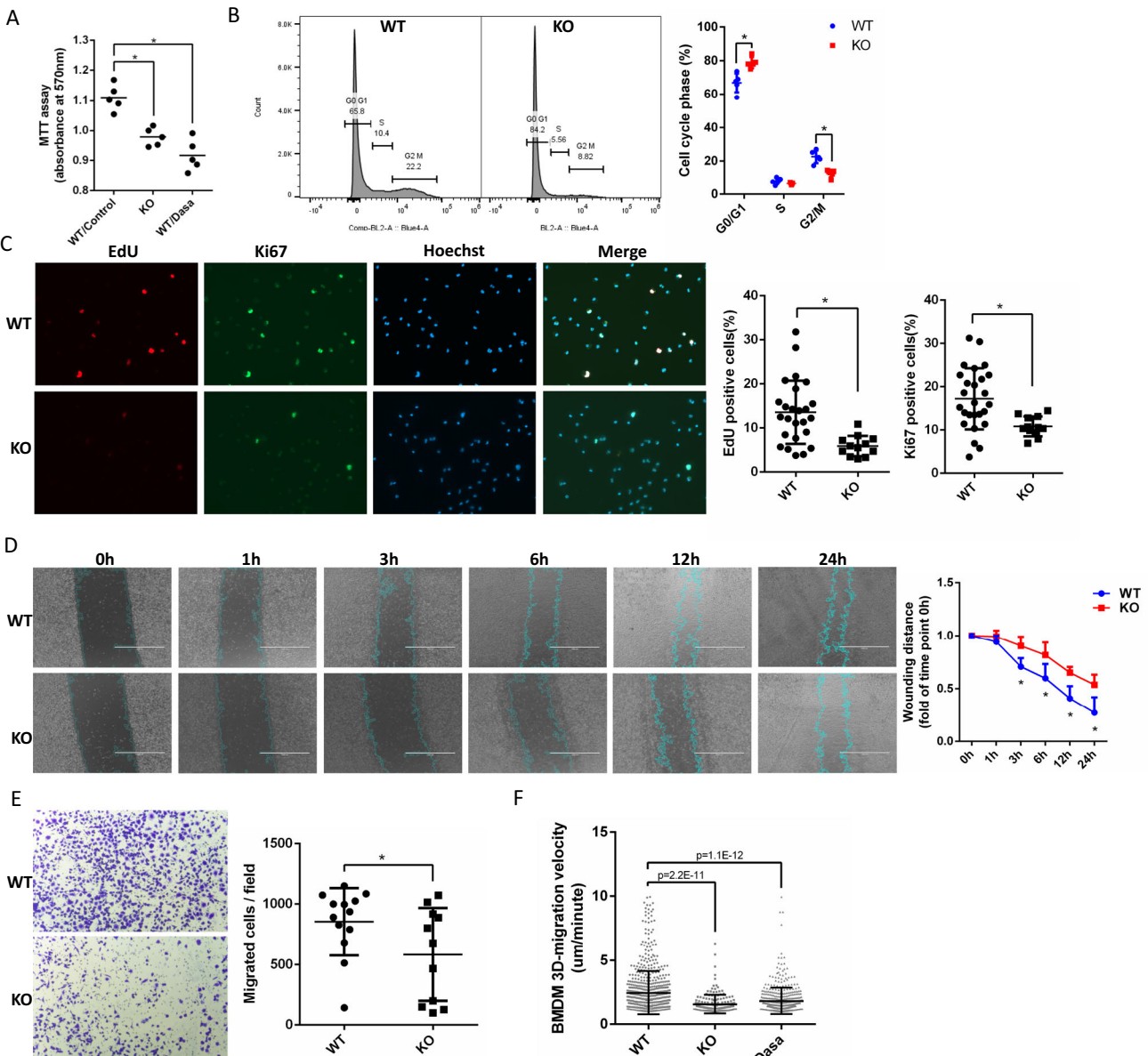

**Fig. 3 | Inhibiting HCK activity and HCK KO decreased macrophage proliferation, and migration in BMDM and Raw264.7. A** BMDM cells of WT and HCK KO and dasatinib treatment were tested with MTT assay for cell proliferation. **B** Representative image and quantification of propidium Iodide (PI) stain detected with flow cytometric to measure cell cycle for WT and HCK KO BMDM. **C** Click-iT™ Edu cell proliferation assay and Ki67 IF staining were performed to measure WT and HCK KO BMDM proliferating. **D** Scratch assay was performed in BMDM cells for WT and HCK KO to measure cell migration. The blue lines are the scratch edge of the cells generated by the software when to measure the scratch distance. **E** 3D transwell assay with Matrigel® Matrix to measure cell migration for WT and HCK KO in BMDMs. **F** 3D-random migration analysis was performed with long time live cell imaging for BMDMs with WT, HCK KO and dasatinib treatment. (**A**–**E**) *$p < 0.05$. Source data are provided as a Source Data file.

towards M2 phenotype (Fig. 5G). We also found that proliferation of macrophage decreased in HCK KO mice in UUO kidneys with co-IF staining of EdU and F4/80 (Fig. 5H). These data suggested that HCK KO attenuates fibrosis by effects on macrophage number as well as phenotype by reduced pro-inflammatory M1-like polarization of infiltrating macrophages in our UUO model. These data also confirmed a critical role for macrophage HCK in regulation of activation, proliferation and polarization and in turn, the progression of renal fibrosis.

### Knockout of HCK in macrophages attenuates renal fibrosis and inflammation in uIRIx mouse model

Next, we studied the role of HCK in unilateral ischemia reperfusion injury with contralateral nephrectomy (uIRIx) model, a preferred model for study macrophage infiltration and function[24,25] (Fig. 6A). At 28 days post-surgery we assayed macrophage infiltration, polarization and kidney fibrosis. In the uIRIx mice, KO of HCK resulted in reduced urine ACR and serum BUN as compared to WT mice (Fig. 6B). By hematoxylin and eosin (H&E) staining, we found that HCK KO mice had reduced tubulointerstitial fibrosis by using Masson trichrome and ColA1-IF staining in the uIRIx kidneys (Fig. 6C). Consistent with the findings in the UUO mice, macrophage infiltration in HCK-KO uIRIx kidneys was significantly decreased (Fig. 6D). Western blot demonstrated that both M1 (iNOS) and M2 (CD206) markers decreased in HCK KO uIRIx kidneys due to the reduction of total macrophage number (Fig. 6E). Similarly, when adjusted for macrophage numbers, M1 marker iNOS was decreased and M2 marker CD206 increased in the

**Table 1 | Top-ranked proteins identified by immunoprecipitation-mass spectrometry (IP-MS) that interact with HCK, FYN and SRC**

| Identified proteins (for HCK) | Coverage | calc. pI | Spectra# | Spectra# | Ratio |
|---|---|---|---|---|---|
| *Tyrosine-protein kinase (HCK)* | 72.62357 | 6.7 | 14 | 389 | 24.44 |
| E3 ubiquitin-protein ligase CBL (CBL) | 15.67329 | 6.54 | 0 | 10 | 6.00 |
| Autophagy-related protein 2 homolog A (ATG2A) | 5.159959 | 5.88 | 0 | 8 | 5.00 |
| Neural Wiskott-Aldrich syndrome protein (WASL) | 20.19802 | 7.93 | 0 | 6 | 4.00 |
| Guanine nucleotide-binding protein b4 (GNB4) | 32.05882 | 6 | 1 | 10 | 4.00 |
| Ras-related protein Rab-3D (RAB3D) | 25.57078 | 4.93 | 0 | 6 | 4.00 |
| *Identified proteins (for FYN)* | *Coverage* | *calc. pI* | *Spectra#* | *Spectra#* | *Ratio* |
| Tyrosine-protein kinase Fyn (FYN) | 54.93482 | 6.67 | 1 | 71 | 24.33 |
| Hsp90 co-chaperone Cdc37 (CDC37) | 30.68783 | 5.25 | 0 | 18 | 10.00 |
| Protein scribble homolog (SCRIB) | 10.2719 | 5.1 | 0 | 11 | 6.50 |
| Fatty acid synthase (FASN) | 14.01832 | 6.44 | 1 | 14 | 5.33 |
| ATP-dependent RNA helicase A (DHX9) | 18.34646 | 6.84 | 0 | 8 | 5.00 |
| Heat shock protein HSP 90-alpha (HSP90AA1) | 50.81967 | 5.02 | 9 | 53 | 5.00 |
| *Identified Proteins (for SRC)* | *Coverage* | *calc. pI* | *Spectra#* | *Spectra#* | *Ratio* |
| Proto-oncogene tyrosine-protein kinase Src (SRC) | 71.64179 | 7.42 | 6 | 1238.00 | 155.00 |
| V-type proton ATPase catalytic A (ATP6V1A) | 75.20259 | 5.52 | 0 | 96.00 | 49.00 |
| DNA-dependent protein kinase catalytic (PRKDC) | 34.4719 | 7.12 | 2 | 141.00 | 35.75 |
| V-type proton ATPase subunit B (ATP6V1B2) | 65.55773 | 5.81 | 0 | 68.00 | 35.00 |
| ATP-binding cassette sub-family D 3 (ABCD3) | 44.00607 | 9.36 | 0 | 37.00 | 19.50 |
| Probable ATP-depen RNA helicase (DDX20) | 35.43689 | 6.95 | 0 | 36 | 19.00 |

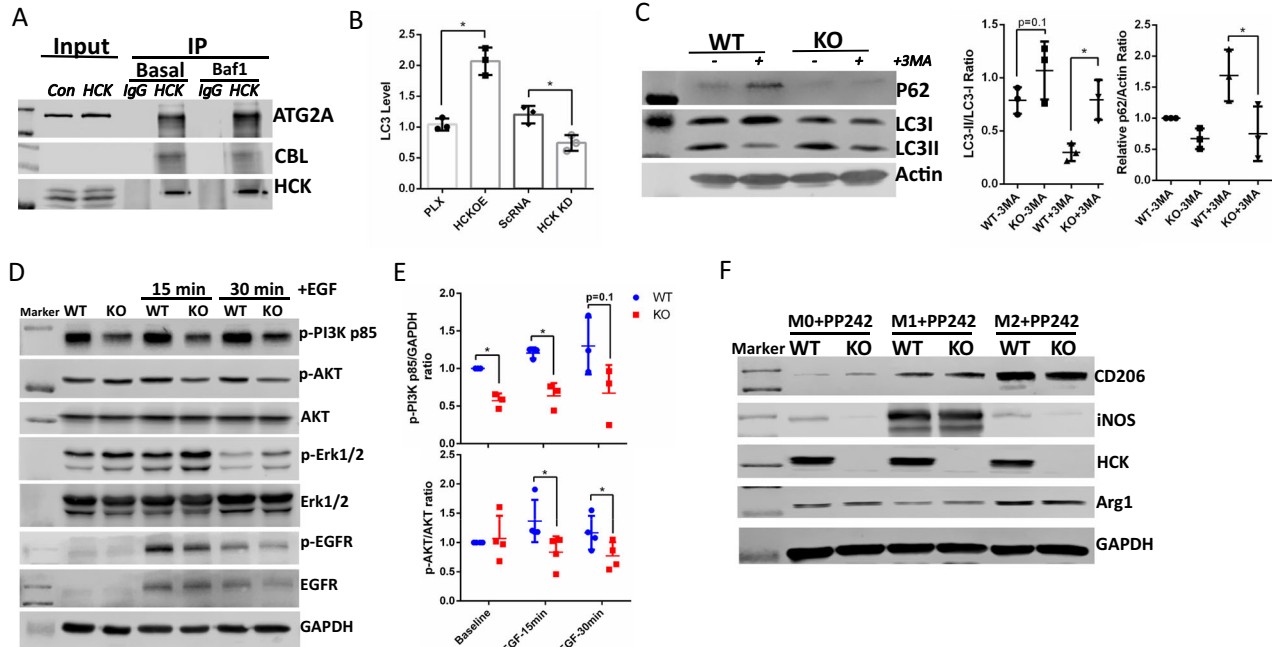

**Fig. 4 | HCK regulated macrophage polarization through autophagy.**
**A** Endogenous HCK interacted with autophagy proteins ATG2A and CBL in macrophage cell line Raw264.7 by IP/WB. **B** Overexpression of HCK inhibited autophagy flux as more LC3 HiBiT reporter remained. HCK knockdown promoted autophagy flux as less LC3 HiBiT reporter remained. **C** HCK KO could promote autophagy by increasing LC3II/LC3I and decrease P62 levels with and without 3-MA treatment in mice BMDM cells. Western blots (**D**) and quantification (**E**) showed phospho-PI3K and phospho-AKT decreased in HCK KO BMDMs. However, there were no significant differences for phospho-Erk1/2 and phospho-EGFR with HCK KO. **F** Effect of HCK KO was abrogated in autophagy inducing by PP242 treatment. Baf1: bafilomycin A1. 3-MA: 3-methyladenine. Source data are provided as a Source Data file.

KO IRI kidneys indicating that HCK KO could inhibit macrophage inflammatory M1 polarization and increase M2 polarization (Fig. 6E). Autophagy activity was increased in macrophages from the uIRIx kidneys of HCK KO mice as reflected by co-staining LC3 and F4/80 and quantification of LC3 in macrophages (Fig. 6F). These data confirmed that HCK KO attenuates fibrosis by reduced macrophage infiltration and altered macrophage phenotype with marked effects on inflammatory M1 polarization in our uIRIx model.

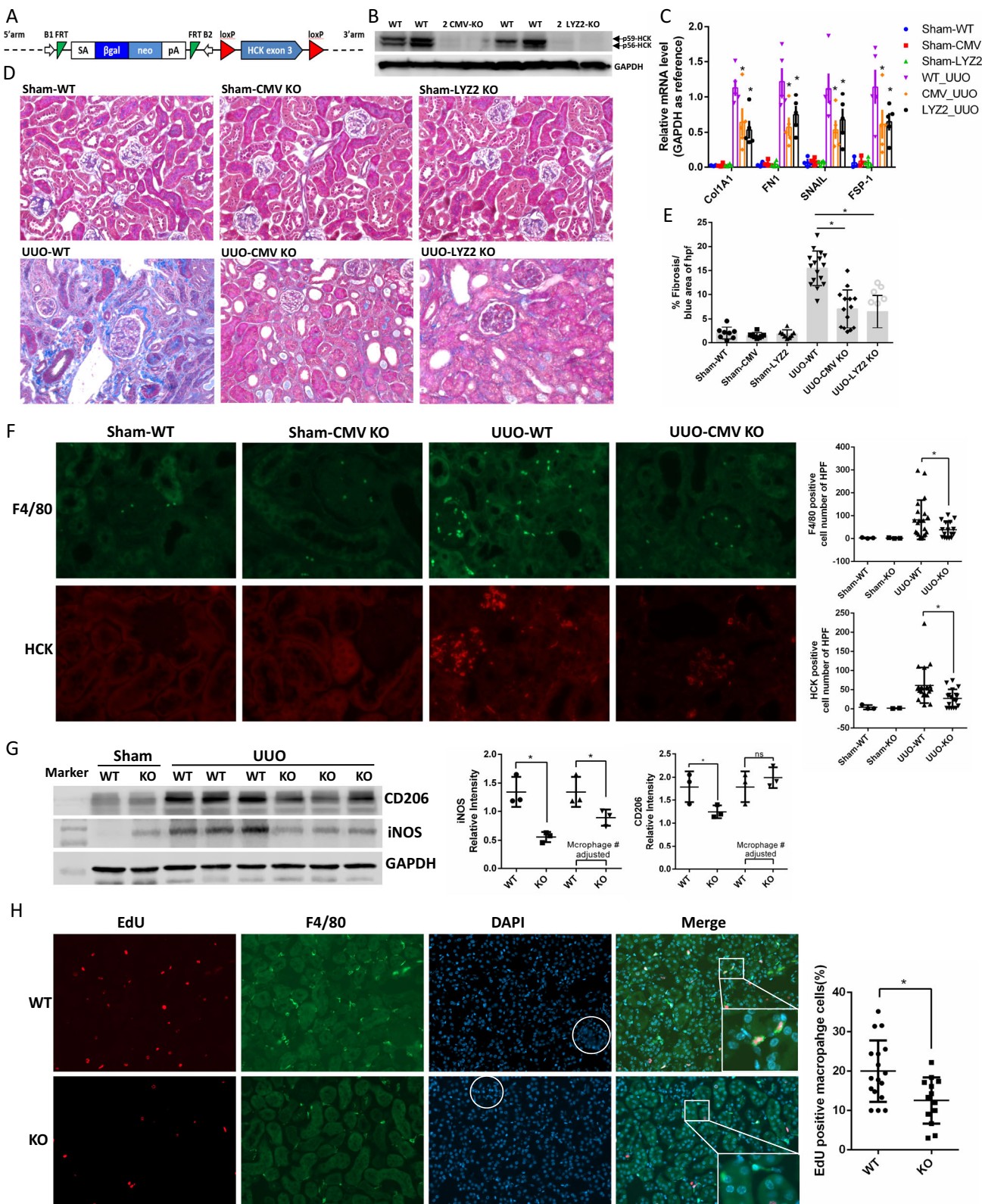

**Fig. 5 | HCK knockout globally and in myeloid cell ameliorated kidney fibrosis in a murine UUO model. A** Schematic diagram shows the strategy of loxP-HCK knock-out mice developed in EuMMCR. **B** Western blot demonstrated HCK global (CMV) and myeloid cell specific (LYZ2) KO in BMDM from mice. **C** mRNA levels of pro-fibrotic markers were decreased in HCK global and myeloid cell KO UUO kidneys compared to WT mice. **D** Representative Masson's trichrome staining images from sham operated & UUO kidneys of WT, CMV KO and LYZ2 KO mice at 7 days post-UUO. **E** Morphometric quantification (*n* = 5 animals; 5 random fields/animal) of the fibrosis positive area for Masson stain. **F** Immunofluorescence (IF) stain indicated F4/80 positive macrophage and HCK both dramatically decreased in HCK KO UUO kidneys. **G** Western blot to show M1 and M2 macrophage markers and quantification of band intensity and adjusted by macrophage number in UUO kidneys. **H** Representative images and quantification (*n* = 5 animals; 5 random fields/animal) of IF staining for EdU positive F4/80 macrophages in UUO kidneys. (**C**, **E**, **F**, **G** and **H**) *p* < 0.05. Source data are provided as a Source Data file.

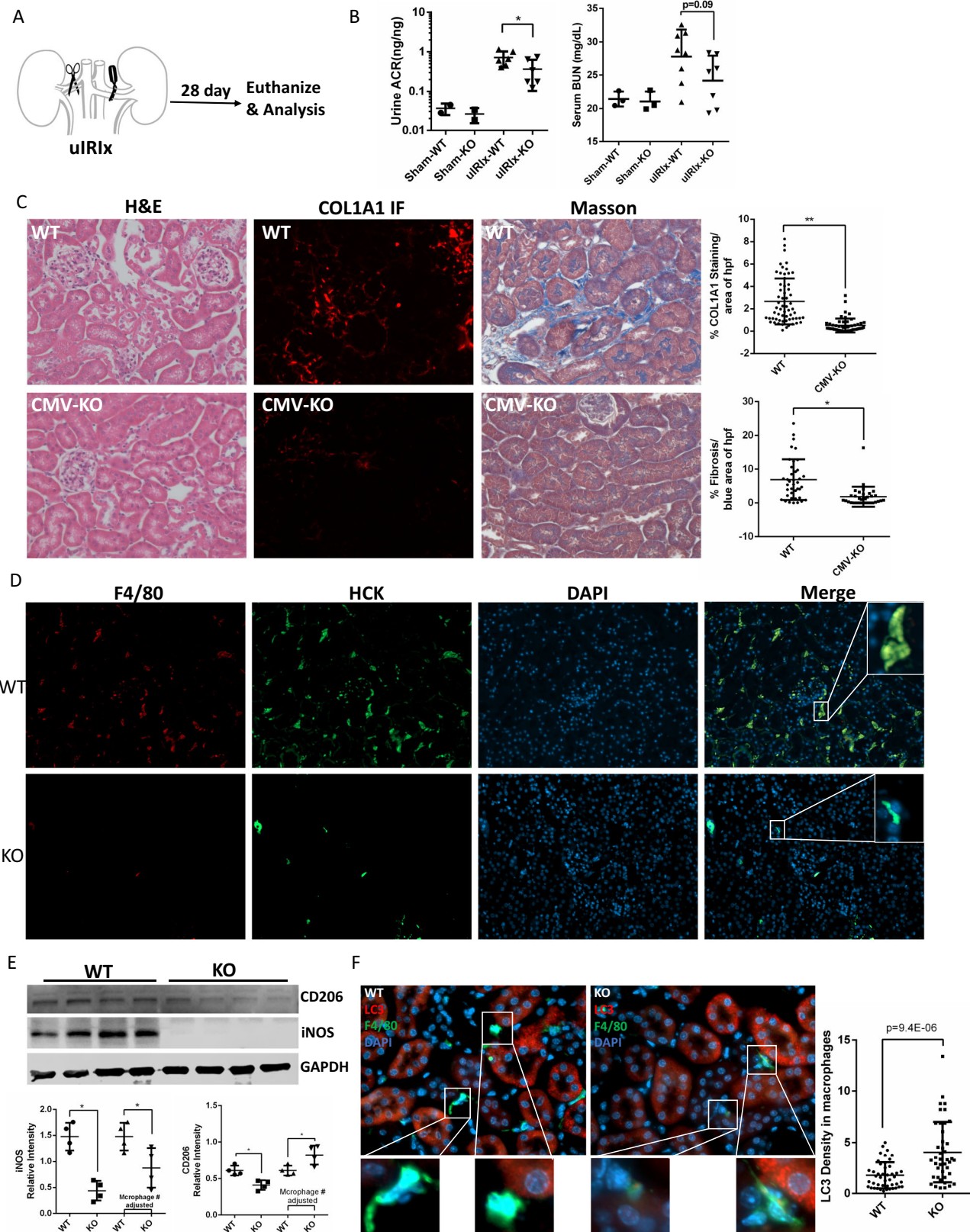

**Fig. 6 | HCK knockout reduced kidney fibrosis in uIRIx model with regulation of macrophage activation. A** Schematic diagram of strategy of uIRIx models. **B** Urine ACR and serum BUN for WT and HCK KO uIRIx mice. **C** Representative images of H&E, COL1A1 IF and Masson trichrome staining from WT and KO uIRIx kidneys. Morphometric quantification (*n* = 5 animals; 5 random fields/animal) of the COL1A1 (top) and Masson (lower) positive area. **D** Representative IF staining images and quantification of from kidneys of HCK and F4/80 macrophage from WT and HCK KO uIRIx kidneys. **E** Western blot and quantification of M1 and M2 macrophage markers from WT and HCK KO mice kidneys at 28 days post uIRIx. **F** Representative IF staining images and quantification of LC3 in macrophages from WT & HCK KO uIRIx kidneys. **B**, **C** and **E** *p* < 0.05 with *t* test. Source data are provided as a Source Data file.

## Development of boron based pharmacophore group for HCK inhibitors with SAR studies of dasatinib

Our previous studies showed that dasatinib at low dose (10 mg/kg) attenuated kidney fibrosis in mice with UUO and lupus nephritis. However, dasatinib at high dose (20 mg/kg) caused proteinuria in lupus nephritis mice and podocytes foot process effacement[26] (Supplementary Fig. S3). The dasatinib renal toxicity was caused by podocyte toxicity due to the low specificity of dasatinib[26]. Therefore, we performed structure activity relationship (SAR)[27,28] studies of dasatinib and screened selective HCK inhibitors by comparing the binding affinity of HCK with other SRC family kinases. We designed and synthesised 33 boron based new pharmacophore groups as HCK inhibitors (Fig. 7A, Supplementary Fig. S4, S5). From Dasatinib we synthesized BT-294 lead compound (by changing thiazol and pyrimidinyl group of dasatinib in square to triazolo-thiadiazine derivative), then from BT294 we synthesized BT332(Thiazole amide derivatives), then we synthesized BT442 lead compound, then at end BT424 (oxadiazaborole derivatives). We found that BT424 had more selective inhibition of HCK at EC50 of 12 uM as compared to other SFKs (Fig. 7B, C, D). We performed cytotoxicity studies of BT424 and did not reveal any cytotoxicity at 50 uM for RAW264.7 macrophages and podocytes (Fig. 7E, F). We performed stability test for BT424 by diluting 100 mM stock in DMSO to 25 uM with water and keep it at room temperature for 24 h, HPLC assay didn't find degradation (less than 3%). These data indicate that we developed HCK more selected inhibitor BT424 and could be our target-to-hit compound.

## HCK inhibitor BT424 decreased macrophage M1 polarization, proliferation and migration

Firstly, we tested BT424 effects on autophagy in BMDMs. BT424 is insoluble in water, we made the stock in DMSO with 100 mM and diluted to 25 uM in medium to treat cells. We found that BT424 treatment could increase autophagy activity with higher LC3II/LC3I ratio and lower P62 levels (Fig. 8A). Inhibition of HCK activation with BT424 decreased macrophage M1 pro-inflammatory polarization and increased M2 polarization in BMDM as shown by western blot of M1 and M2 markers of iNOS and CD206 (Fig. 8B). Next, we measured the effects of BT424 on macrophage proliferation. We first performed MTT assay in Raw264.7 (Fig. 8C) cells and found that BT424 significantly decreased proliferation of these cells. We also used propidium iodide (PI) in Raw264.7 to measure cell cycle and found BT424 treatment increased G0/G1 phases and decreased G2/M phases (Fig. 8D). Click-iT™ EdU cell proliferation assay (Fig. 8E) and Ki67 IF staining (Fig. 8F) were performed in Raw264.7. EdU and Ki67 positive cells were decreased in BT424 treatment, compared to control cells. We performed scratch assay in Raw264.7 cells and found that BT424 significantly decreased migration of these cells (Fig. 8G). These data demonstrated that by inhibiting HCK activity BT424 could inhibit macrophage M1 pro-inflammatory polarization, proliferation and migration.

## HCK specific inhibitor BT424 ameliorate inflammation and kidney fibrosis in UUO model

We first tested the toxicity of our HCK inhibitor by daily gavage in WT mice with BT424 at 25 mg/kg for 1 month. We did not observe any obvious toxicity based on the body weight changes, behavior, and physical activity of the mice. In addition, we performed Routine Chemistry Panel test from IDEXX BioAnalytics (Westbrook, ME 04092) with the serum from vehicle- and BT424-treated mice and did not see noticeable changes in liver enzymes, renal function, lipid profile, and glucose levels (Supplementary Table S1). These data indicate that treatment of BT424 does not induce significant toxicity in these mice. We then tested the effect of BT424 in UUO-induced fibrosis. The mice were treated with BT424 at 25 mg/kg or vehicle (n = 5) by daily gavage starting one day prior to the surgery to 7 days post-surgery. In H&E staining, we found that BT424 treatment reduced morphologic tubular

injury in the UUO kidney vs vehicle-treated mice (Fig. 9A). Masson and Col1A1-IF staining confirmed that BT424 treatment reduced renal fibrosis in the UUO kidneys (Fig. 9A, B). Col1a1, Fibronectin and FSP-1 mRNA expression were significantly decreased in the UUO kidney of BT424-treated mice (Fig. 9C). F4/80-positive macrophage population was significantly decreased in BT424 treated mice with UUO (Fig. 9D). Western blot demonstrated that iNOS decreased due to the reduction of total macrophage number and reducing M1 polarization (Fig. 9E). CD206 reduced in BT424 treated kidneys due to the reduction of total macrophage number but increased after macrophage number adjustment as BT424 regulating macrophage suggesting skewing from M1 toward M2 polarization (Fig. 9E). Autophagy activity was increased in macrophages from the UUO kidneys of BT424 treated mice as reflected by co-staining LC3 and F4/80 and quantification of LC3 in macrophages (Fig. 9F). These data confirmed that BT424 attenuates fibrosis by reduced macrophage number and reduce macrophage inflammatory M1 polarization through autophagy in UUO model.

In summary, we unravel a mechanism connecting the SFK HCK to progressive kidney IF/TA in CKD and CAI, by regulating autophagy within macrophages, altering their polarization, proliferation, and migration into diseased kidney in response to injury. We also developed a non-toxic specific HCK inhibitor ie a target-to-hit compound BT424, and demonstrated its regulation of macrophage function leading to attenuation of progressive renal fibrosis.

## Discussion

In the current study, we demonstrated that HCK is a key driver of kidney inflammation and fibrosis. Our prior data clearly suggested that the renal expression of HCK increases in patients with native or allograft CKD and associated with adverse outcomes. Here we demonstrate that HCK is expressed predominantly in macrophages within the kidney and show that HCK promotes macrophage M1 polarization in BMDMs and kidney-infiltrating macrophages, and that HCK-inhibition or KO induced phenotypic changes in macrophages including impaired proliferation and migration. These effects of HCK are likely through its direct interaction with autophagy proteins (ATG2A and CBL) which culminate in suppression of autophagy in macrophages. We then confirmed the role of macrophage HCK in renal fibrosis by using macrophage specific HCK knockout mice in renal fibrosis models, with consistent reduced macrophage infiltration in the diseased kidney. In addition, we developed a boron-containing selective HCK inhibitor, which inhibits M1-like polarization, restores autophagy activity in BMDM and kidney-derived macrophages and attenuates renal inflammation and fibrosis in mice with UUO and unilateral ischemia and reperfusion. In summary, HCK induces kidney fibrosis broadly through regulating macrophage function including proliferation, inflammatory polarization and migration.

Macrophage is a master regulator for kidney inflammation and fibrosis[6,29], but conflicting reports suggest both pro- and anti-fibrotic roles of macrophages[6]. Classically activated macrophages (M1) secrete a series of pro-inflammatory factors (IL-1, IL-6, IL-12, TNF-α), chemokines (IL-8), activated oxygen species, and nitric oxide (NO) which promote inflammation and damage of tissues[6,30]. A large body of evidence suggest that M1 macrophages promote kidney injury and fibrosis[31–36]. The role of M2 macrophage in kidney fibrosis is controversial due to its complex functions and are further divided into subclasses. M2a macrophages are mainly anti-inflammatory and promote epithelial healing and regeneration of intact tubules[37,38]. M2c macrophages were thought to promote kidney and liver fibrosis via secretion of TGF-β1[39,40]. However, macrophage-specific deletion of TGF-β1 failed to prevent renal fibrosis in animal models of ischemia-reperfusion or obstructive nephropathy[41]. In contrast, selective deletion of TGF-β receptor II (TβRII) in monocytes/macrophages promoted kidney fibrosis by enhancing renal macrophage infiltration[42]. Hence, the heterogeneity and plasticity of macrophages poses a significant

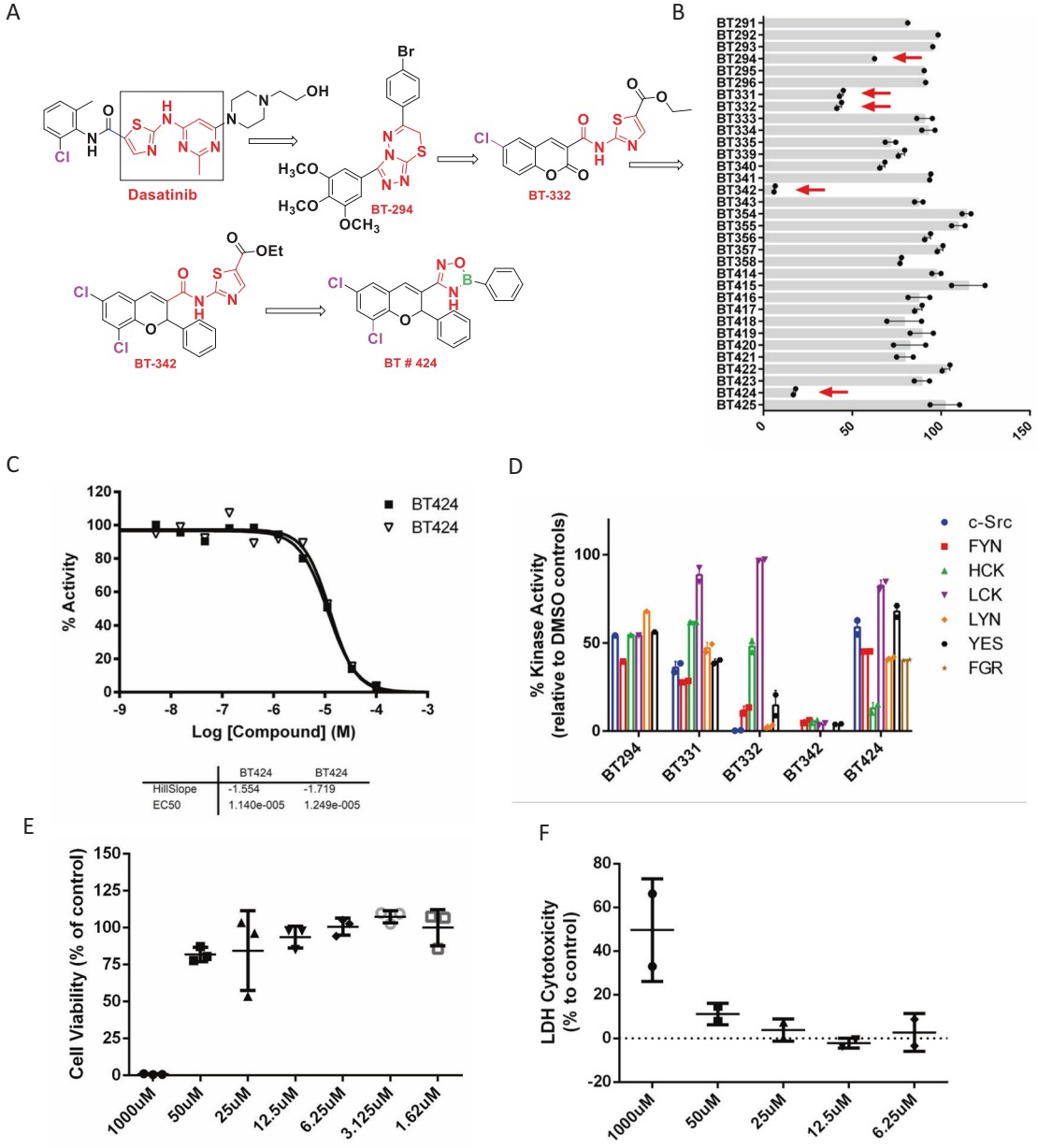

**Fig. 7 | Development of selective HCK inhibitors and identification of BT424 as a specific HCK inhibitor. A** SAR study of dasatinib to design selective HCK inhibitors. **B** Screen HCK specific inhibitors for all our designed compounds. Five inhibitors (BT294, 331, 332, 342 and 424) with high efficiency of HCK inhibitions are highlighted in red arrows. **C** HCK inhibition profiling for BT424 with 10 dose 1:3 serial dilution. **D** Inhibition of all SFKs for 6 candidate HCK inhibitors at 50uM.

SFKs inhabitation assay was performed at Reaction Biology Corp. Values were shown for % of enzyme activity compared to DMSO control. Cell proliferation (**E**) and cytotoxicity (**F**) were measured by MTT assay and release LDH with various concentration of BT424 for 24 h in Raw264.7 cells. Source data are provided as a Source Data file.

obstacle to study its functions and likely explains why depleting macrophages globally has often yielded conflicting results.

The overall role of the HCK in macrophage activity has been an area of much controversy, with conflicting findings in different cells and disease conditions. In the tumor associated macrophages (TAM), Poh et al. showed that HCK's activity is associated with alternatively activated endotype[43]. Similar to this, others demonstrate that genetic ablation or chemical inhibition of HCK reprograms TAM to an inflammatory endotype and enhance T cell recruitment and activation[44–46]. However, HCK has opposite effects in non-cancer cells by promoting macrophage inflammatory activation. Ernst M, et al. generated HCK constitutively active mutant mice and found that these mice developed spontaneous pulmonary inflammation and enhanced

innate immune response[47]. Later, they demonstrated that HCK, FGR, LYN are critical in generation of in vivo inflammatory environment in autoantibody-induced arthritis and other inflammation mouse models[48]. Another study showed that loss of HCK/LYN increases M2 macrophage programming[15]. Our data are consistent with these latter findings, and show that inhibition or KO of HCK reduced pro-inflammatory polarization (M1-like) and increased M2-like polarization of macrophages, associating with reduced kidney injury and fibrosis in these animal models.

Autophagy is recognized to be a highly conserved homeostatic pathway with complex roles in regulating macrophage polarization[21]. It must also be noted that autophagy inhibition has been associated with increased M1 polarization or M2 inhibition in previous data[49–51].

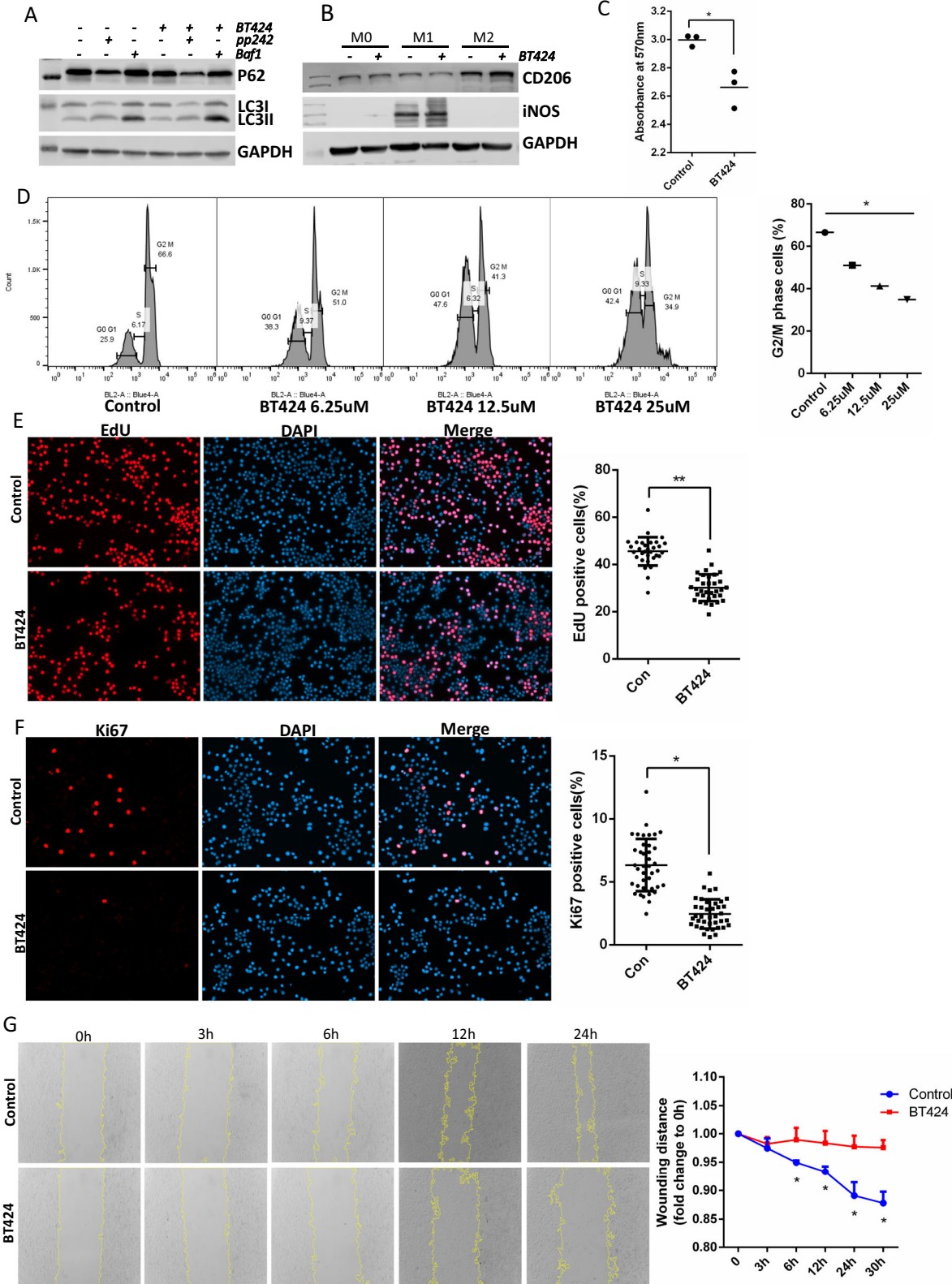

**Fig. 8 | HCK inhibitor BT424 regulated macrophage activation and autophagy in vitro.** Western blot to show BT424 regulation of autophagy (**A**) and macrophage polarization (**B**) in Raw264.7 macrophage cell line. **C** MTT assay was tested for BT424 treatment with gradient dosages for cell proliferation in Raw264.7. **D** Representative of flow cytometric image and quantification for propidium Iodide (PI) stain to measure cell cycle for control and BT424 treatment in Raw264.7 cells.

**E** Click-iT™ EdU cell proliferation assay and Ki67 IF staining (**F**) were performed to measure proliferation of Raw264.7 with control and BT424 treatment. **G** Scratch assay was performed in Raw264.7 cells for control and BT424 treatment. The yellow lines are the scratch edge of the cells generated by the software when to measure the scratch distance. **C**–**G** *$p$ < 0.05, D with ANOVA test, others with $t$ test. Baf1: bafilomycin A1. Source data are provided as a Source Data file.

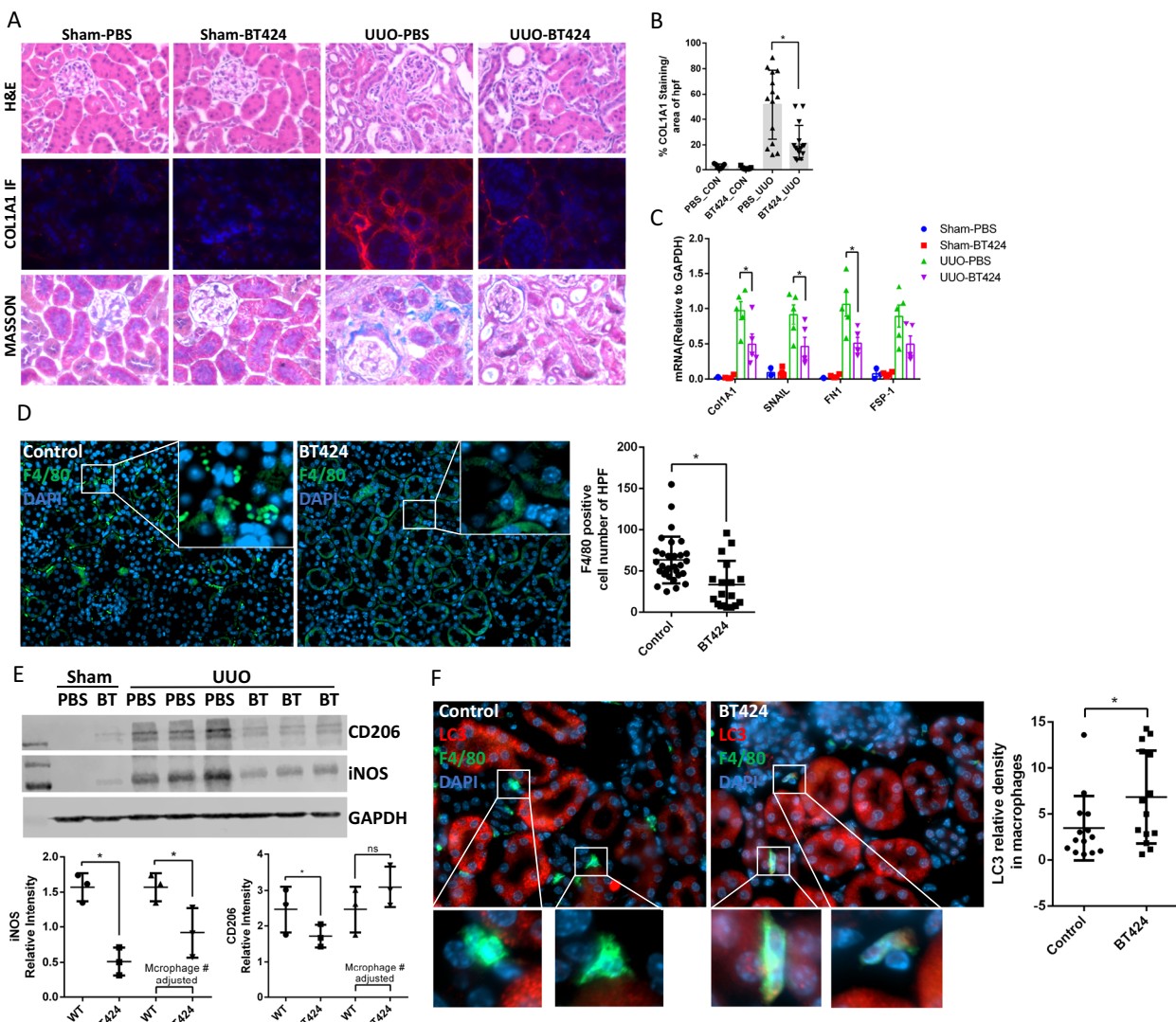

**Fig. 9 | Inhibition of HCK by BT424 reduced renal fibrosis process in UUO model by regulating macrophage activation and autophagy. A** Representative images of H&E, COL1A1 IF and Masson trichrome stain from sham operated & UUO kidneys of BT424- or vehicle-treated mice. **B** Morphometric quantification (*n* = 5 animals; 3-4 random HPFs/animal) of the COL1A1 IF stain positive area. **C** mRNA levels of pro-fibrotic markers at 7 days post-UUO by qPCR. **D** IF stain showed F4/80 positive macrophage dramatically decreased in BT424 treated UUO kidneys. **E** Western blot assay demonstrated macrophage makers in 7 days post-UUO kidneys. Quantification for bands intensity and adjusted by macrophage cell numbers. **F** IF stain showed LC3 intensity was increased in F4/80 positive macrophage in BT424 treated UUO kidneys. **B**–**F** *t* test *$p < 0.05$. Source data are provided as a Source Data file.

Together, these may suggest that regulation of macrophage function by autophagy may be context dependent. Other studies have proposed multiple mechanisms for regulation of autophagy by HCK[52,53]. Here, we demonstrated a interaction between HCK and ATG2A and CBL, that may underlie its regulation of autophagy and show that HCK-KO increases autophagy activation in Raw264.7 and BMDMs. However, the mechanisms of autophagy regulation by HCK are not fully defined. Consistent with the previous findings, we identified that HCK KO in BMDMs decreased PI3K/AKT pathway[52,54], which is a known regulator of autophagy pathway through mTORC1. In addition, HCK likely plays multiple roles in cell signal transduction in macrophages. For instance, our data in BMDM and mouse models indicated HCK regulates macrophage migration. This is potentially through integrin signaling pathway, as we showed a decrease of F-actin spreading, phospho-SYK in HCK-KO cells. Independent studies have also shown that HCK regulates macrophage spreading and three-dimensional migration through integrin signaling pathway in NFS-60, BMDM and other myeloid cells[16,55–57]. Here, we also observed that HCK mediates LPS-induced macrophage M1 polarization, suggesting a role for HCK in

regulating TLR signaling. Consistent with our data, other studies showed that HCK can regulate TLR signaling induced by TNF-a and IL-6[58].

Our studies suggest that among the members of SFK, HCK expresses is highly regulated in the macrophages of diseased kidney. On other hand, FYN expresses mostly in podocytes and regulates podocyte function by phosphorylation of nephrin[59]. We have previously shown that dasatinib induces proteinuria in lupus nephritis mice likely through inhibition of FYN-induced podocyte injury[14]. Therefore, it is critical for us to develop more selective inhibitors of HCK, which will not affect FYN activity. We described here BT424, a relatively selective inhibitor of HCK and BT424 does not affect podocyte injury and therefore, we believe that BT424 is a better drug to target HCK as an anti-inflammatory and anti-fibrosis therapy in patients with kidney disease.

Our study has several limitations. We demonstrated that HCK is the main SFK in macrophages and highly regulated in kidney disease, but we could not rule out the role of other SFK members in kidney disease. We are also aware that macrophages have multiple functions

in kidney disease. Recently, macrophages to myofibroblasts trans-differentiation have also been described[60] and we will test whether this process is also regulated by HCK in our future studies. Also, total F4/80-positive cells, which includes macrophages (infiltrated and resident) but also some dendritic cells[61], were significantly reduced in the HCK KO mice with UUO. It would be specifically interesting to study the crosstalk between macrophages, tubular cells, and fibroblasts in the context of HCK knockout in future work. LysM-cre is not macrophage specific because the LysM promoter also expresses in neutrophils[62]. Previous studies suggest that HCK regulates neutrophil activation[63,64] and migration[65,66]. Therefore, future studies are required to distinguish the roles of HCK in macrophages from neutrophils in kidney inflammation and fibrosis. We used HCK inhibitor BT424 to treat the mice before the injury, indicating the prevention of kidney injury or fibrosis in these mouse models. Future studies are also required to determine whether treatment of the mice after disease onset can reverse the kidney injury and fibrosis.

In summary, we demonstrate a critical role of HCK in regulation of macrophage function in the context of kidney inflammation and fibrosis. Using global- and cell-specific- HCK-KO models and by developing a selective boron containing HCK inhibitor, we demonstrate the therapeutic potential of this pathway in progressive kidney disease.

## Methods

### GoCAR Study

The GoCAR study is a prospective, multicenter study from USA and Australia to investigate the association of differential gene expression and the development of chronic allograft injury. The study was approved by the institutional review boards of the participating institutions. Written informed consent was obtained from all enrolled patients. Details of the observational GoCAR cohort, including ethics approval, eligibility and exclusion criteria are published elsewhere[17].

### Cells

RAW264.7 (TIB-71™), HEK293 (CRL-1573™) and L929 fibroblasts (CCL-1™) cells commercially obtained from ATCC. HEK293, RAW264.7 and L929 cells were expanded using DMEM (Gibco) media with 10% FBS and 1% Pen/Strep. Bone marrow-derived macrophages (BMDM) were isolated from mice following the protocols[67,68] and cultured using full DMEM (Gibco) with 20% of L929 cell supernatant for 7 days. For M1 activation, 100 ng/ml LPS with 50 ng/ml IFNγ was added in culture medium; for M2 activation, 10 ng/ml IL-4 was used to treat for 24 h.

### Compounds

Dasatinib was purchased from Selleckchem (Cat#: S1021) and was made stock at 10 mM at DMSO, then it was diluted to 100 nM to treat R264.7 and BMDM. Structure activity relationship (SAR) studies[27,28] was performed by Dr. Bhaskar Das. BT424 was synthesized by Dr. Das's lab and purified with HPLC after synthesis. A step-by-step experimental procedure of BT424 synthesis and its chemical characterization were provided in the supplementary Fig. S4. BT424 is insoluble in water. It was stocked at 100 mM in DMSO and diluted to 25 uM to treat cells.

### Plasmids and transduction

Plasmids were purchased from Dharmacon (https://horizondiscovery.com/) for HCK, FYN and SRC overexpression and shRNA knockdown. HCK overexpression (Clone ID: ccsbBroad304_00733) and knockdown (Clone ID: V2LHS_237454, V2LHS_133010, V2LHS_133007). FYN overexpression (Clone ID: ccsbBroad304_00600), SRC overexpression (Clone ID: ccsbBroad304_06992). Plasmids for control (for OVE, Cat#: OHS6085, for KD, Cat#: RHS4349). Transfecting in Raw264.7 was used Lipofectamine LTX Reagent (Cat #: A12621, Thermo Fisher Scientific) and in HEK293 used PolyJet™ (SignaGen Laboratories, Cat #: SL100688) following manufacturer's instructions.

### Quantitative PCR

RNA was extracted using the TRIzol Reagent (Thermo Fisher Scientific) from cell and whole kidneys. One thousand nanograms of RNA was used to prepare cDNA. Quantitative real-time PCR was performed using Qiagen QuantiTect One Step RTPCR SYBR green kit (Qiagen). Data were analyzed by the $2-\Delta\Delta CT$ method and presented as fold change relative to a control sample after normalization against the expression of housekeeping genes.

### Western blotting

Cells were lysed with Cell Lysis Buffer II (ThermoFisher Scientific, # FNN0021) containing 1% NP-40, added protease inhibitor cocktail (Sigma, #P1860), L-isozymes of alkaline phosphatase inhibitor cocktail (Sigma, #P2850) and tyrosine and serine/threonine phosphatase inhibitors (Sigma, #P5726). Lysates were subjected to immunoblot analysis using the following antibodies: HCK (CST #14643), V5 tag (GenScript Inc #A01724), phosphor-Y410 HCK (Abcam #ab61055), LC3A/B (CST #4108), Fyn (CST #4023 S), p62/SQSTM1 (NOVUS BIOLOGICALS NBP1-48320SS), Atg2A (CST #15011), c-Cbl (CST #2747), phospho-c-Cbl (Tyr700) (D16D7) (CST #8869), MMR/CD206 (R&D Systems #AF2535), iNOS (BioLegend #690902), Arginase 1 (BioLegend #678802), Phospho-PI3K p85 /p55 (CST #17366), Phospho-Akt (CST #4060), Akt (CST #4691), Phospho-p44/42 MAPK (Erk1/2) (CST #9101), p44/42 MAPK (Erk1/2) (CST #9102), EGFR (CST #4267), Phospho-EGFR (CST #3777), SYK (CST #13198), pY20 (Invitrogen # 14-5001-82), a-SMA (Sigma-Aldrich #A5228) and GAPDH mAb (CST #2118). Anti-V5-tag mAb-Magnetic Beads (MBL #M167-11) were used for immunoprecipitation. Antibody anti-GAPDH was diluted 5000 times with 3% BSA in PBST. All other antibodies mentioned above for WB were used dilution of 1:1000. The Anti-V5-tag mAb-Magnetic Beads were used 50 ul for each reaction with 400 ul incubation volume. The secondary antibodies were from Promega: Anti-Rabbit IgG (H+L), HRP Conjugate (#W4011) and Anti-Mouse IgG (H + L), HRP Conjugate (#W4021). The secondary antibodies were diluted with 1:5000. Image Studio Lite Ver5.2 for western blot image acquisition. Densitometry was performed on images of Western blots using Image J software.

### Autophagy measurement

Autophagy LC3 HiBiT reporter (Promega, #GA2550) was transfected to HEK293 cells with PloyJet (SignaGen Laboratories, #SL100688) following the manufacture's protocol. Then LC3 reporter was detected with Nano-Glo HiBiT fluorescent reagent (Promega, N3030). The fluorescent signal was adjusted by load protein amount. For Raw264.7 and BMDM, autophagy inducer PP242 at 1 uM, autophagosome degradation inhibitor Bafilomycin A1 (BafA1) at 100 nM, and autophagy inhibitor 3-Methyladenine (3MA) at 5 mM treated for 24 h for LC3 and P62 measurement with WB. HCK inhibitors BT424 and dasatinib were added to pretreat cells 2 h before autophagy stimulation.

### EGFR and integrin signaling pathway studies

BMDMs from WT and HCK KO mice were starving overnight with 1% FBS DMEM with L929 on day 5. Then EGF at 100 ng/ml was used to treat cells for 15 and 30 min before harvesting on ice with cold PBS wash. The cell lysate was then used for western blot assay to measure EGFR signaling-related proteins. For the integrin signaling, BMDMs were plated to collagen pre-coated 10 cm dishes for 1 h following the paper[16]. F-actin and pY20 with SYK were measured by IF stain and western blot.

### Immunofluorescence staining

Immunofluorescence (IF) staining follow our previous paper[14]. In brief, Raw264.7 and BMDMs were cultured on cover glasses and washed with PBS 3 times. Then fixed with 4% PFA for 10 min and blocking for 1 h. Primary antibody with 1:100 dilution incubated overnight, and the next day added fluorescence secondary antibodies. For the UUO and uIRIx

mouse tissue staining, paraffin embedding tissue was used, antigen retrieval was performed first with steamer. The following antibodies were used: anti-phospho Y410 HCK antibody (Abcam, ab61055), anti-HCK antibody (Abcam, ab75839 or Cell Signaling Technology, #14643), anti-LC3A/B antibody (Cell Signaling Technology, #4108), anti-Ki67 antibody (Abcam, ab16667), anti-F4/80 antibody (Invitrogen, 14-4801-82), anti-COL1A1 antibody (Southern Biotech, #1310-01), anti-F-actin antibody (Novus Biologicals, #NB100-64792). The secondary antibodies were from Invitrogen: Goat anti-Rat IgG (H+L) Cross-Adsorbed Secondary Antibody, Alexa Fluor™ 568, Goat anti-Rabbit IgG (H+L) Cross-Adsorbed Secondary Antibody, Alexa Fluor™ 568, Goat anti-Rabbit IgG (H+L) Highly Cross-Adsorbed Secondary Antibody, Alexa Fluor™ 488, Goat anti-Mouse IgG (H+L) Cross-Adsorbed Secondary Antibody, Alexa Fluor™ 488, Goat anti-Mouse IgG (H+L) Cross-Adsorbed Secondary Antibody, Alexa Fluor™ 568. The secondary antibodies were diluted with 1:200. AxioImager. Z2(M) and ZEN 2.3 were used for IHC IF microscopy image acquisition.

HCK and SRC family kinases activity was measured by Reaction Biology Corp, PA.

## Chemokine array

The chemical array was performed with cultured medium from BMDM from WT and HCK KO mice using the Proteome Profiler Mouse Chemokine Array Kit (R&D system, ARY006) following the manufacturer's instructions. BMDMs at 7-day were polarized to M1 with 100 ng/ml LPS with 50 ng/ml IFNγ for 24 h, WT BMDM were pre-treated with HCK inhibitor dasatinib for 2 h before M1 polarization.

## Mass spectrometry

HCK-V5, FYN-V5 and SRC-V5 plasmids were overexpressed in HEK293 cells. Protein lysates after 48 h transfection was immunoprecipitated with anti-V5-tag magnet beads and run with PAGE gels. Three resultant lanes were sent for mass spectrometry (Center for Advanced Proteomics Research, RUTGERS New Jersey Medical School). Proteins identified in control and HCK, FYN and SRC overexpression lanes were selected for further filtering analysis. Proteins in overexpression samples with Spectra # <2 were omitted and ratio of overexpression and control lanes were ranked for identified interacting proteins.

## Click-iT™ EdU cell proliferation assay

Macrophages proliferation was measured with different assays with BMDMs, Raw264.7 and in mice. Click-iT™ EdU Cell Proliferation Kit from ThermoFisher Scientific (#C10340) was used to detect proliferating cells following the manufacture protocol. EdU (5-ethynyl-2′-deoxyuridine) was intraperitoneally injected in mouse (40 mg/kg) or treated BMDM and RAW264.7 (10uM) for 2 h for incorporated into newly synthesized DNA. Kidney tissue and cells were fixed with 4% formaldehyde and coupling of EdU with Alexa Fluor™ 647 dye in Click-iT™ kit following the manufacturer's instructions. For the UUO and uIRIx mouse tissue staining, frozen tissue was used following the manufacture's protocol like cell staining.

## MTT and propidium iodide flow for cell proliferation

BMDMs cultured for 4 days were used for MTT and PI flow assay. eBioscience™ Propidium Iodide staining solution (#00-6990-50) and MTT (#M2003, Sigma) were used and following the manufactory's protocols. Attune Flow Cytometer and FlowJo v10.8.0 were used for flow data analysis. BT424 treated Raw 264.7 for 24 h before the cell proliferation assay.

## 2D and 3D migration assay

Scratch Assays. RAW 264.7 cells and BMDMs were plated on 6 well plates one day prior to the assay day and starved overnight. Starving medium with 0.5% FBS was used to reduce cell proliferation. Scratches were made by using 200 ul pipet tips. Starving medium with 0.5 FBS

was replaced for full culture medium to reduce the cell proliferation. Images were taken at 0 h, 1 h, 3 h, 6 h, 12 h and 24 h after the scratch. The distance of unhealed area was measured with ImageJ package "Wound_healing_size_tool".

Transwell assay. 2D transwell migration assays were performed in 24-transwells chambers (5um pores, Corning, Cat #: 3421) according to the manufacturer's instructions. Briefly, BMDMs were starved for 3 h and then seeded 1E5 cells in the upper chambers. The lower chamber was filled with 750 ul of full DMEM medium with 20% of L929 medium. After 24 h, cells in the upper chambers were removed with cotton swab. Cells on the lower side of the insert membrane were fixed with 4% glutaraldehyde for 10 min at room temperature. Then 0.1% crystal violet was used to stain the cells for 20 min at room temperature following two times of PBS washing. Images were taken after microscope after the insert completely dry. Image J was used to analysis the cell number.

3D transwell assay. The transwell assay was performed following the paper[16]. Briefly, 100 μl of Matrigel diluted to 0.4 mg/ml (Corning® Matrigel® Matrix, Cat #: 354277) was poured into 24 transwell (8 μm pores) and polymerized in cell incubator for 30 min. 700 μl of Full DMEM medium with 20% of L929 medium was put to lower chamber to hydrate the Matrigel and the membrane for 3 h. BMDMs (2E5) in starving medium (0.5% FBS) were seeded in the upper chamber and after 24 h of migration. Cells passed through the transwells were counted as in regular transwell assay above.

3D-random migration analysis was performed following the paper[16]. Briefly, BMDMs (1E5) suspended in Matrigel (Corning, Cat #: 354277) at 4 °C were plated in ibidi 8 well high glass bottom chamber (ibidi, Cat.No:80807) and polymerized in cell incubator for 30 min. 200 ul of full DMEM medium with 15% of L929 supernatant was added to each chamber and kept in 37 °C in 5% CO2 atmosphere to record cell migration. 200X objective of an automated Leica DMIRBE microscope equipped with a CoolSnapEz Camera (Roper Scientific) acquired images every 3 min for 10 h with software LAS X 3.7.4 to record cells migration. The Z stacks imaging was used between 1500 and 2000 μm (30 μm intervals) above the glass surface. Migrating cells were tracked from projected stacks to analyze velocity and persistence over time with ImageJ software TrackMate plugins. The first 15 images were removed to avoid light variation and cell recovery at beginning, and 8-bit image and lower and upper threshold levels were set as 20 and 120. Laplacian of Gaussian (LoG) detector was used. Detailed parameters were set as: Estimated object diameter: 40-micron, Quality threshold: 0.02, Initial thresholding: 0.03. Simple LAP tracker was used to track the cell movement. Parameters were set as: Linking max distance: 30-micron, Gap-closing max distance: 30-micron, Gap-closing max frame gap: 2, number of spots in track: 12.28. TrackMate output files were opened in MATLAB for velocity analysis and AVI format of video files were saved for cell movement tracking. The cells that moved less than 1 um/min were considered as dead cells and omitted.

## Animal studies

HCK exon3 loxp flanked KO mouse were developed at EuMMCR in Germany (HCK ES Cell Clone: HEPD0510). These mice were crossed with tissue specific Cre mice (CMV-cre B6.C-Tg (CMV-cre)1Cgn/J006054; LysM-cre B6.129P2-Lyz2tm1(cre)Ifo/J, 004781, the Jackson Laboratory) to generate HCK KO specifically in tubular cells and macrophages. C57BL/6 J WT mice also from Jackson Laboratory. All mice were maintained in our animal facility (Center for Comparative Medicine and Surgery, CCMS) at Mount Sinai under controlled environmental conditions: 12/12 light/dark cycle, ambient temperature 20–25 °C. More husbandry operation information can be found through CCMS website: https://icahn.mssm.edu/research/ccms/services-rates/husbandry. All mouse experiments were performed per the guidelines and were approved by the Institutional Animal Care and Use Committee at the Icahn School of Medicine at Mount Sinai (New York, NY).

UUO model in WT, HCK KO, and HCK inhibitor mice was performed following our previous paper[14]. BT424 was made in 250 mg/ml in DMSO as stock, then was diluted to 2.5 mg/ml with PBS before gavage mice for final concentration of 25 mg/kg body weight. Unilateral IRI with contralateral nephrectomy (uIRIx) model was performed following the papers[69,70], briefly, artery and vein of right kidney was tied, and the right kidney was removed. Then the left kidney's artery and vein were clamped for 25 min and released. At the end of mice models, the mice were IP injected with 100 mg/kg ketamine and 10 mg/kg xylazine and then perfused with cold PBS for tissue collection.

## Statistics

Data were expressed as mean ± SEM (X ± SEM). 2-tailed unpaired *t* test was used to analyze data between two groups after determination of data distribution. The ANOVA test was used when more than two groups were present. Statistical significance was considered when $P < 0.05$. All statistical analyses and figures were performed using GraphPad Prism software (version 6).

## Reporting summary

Further information on research design is available in the Nature Portfolio Reporting Summary linked to this article.

## Data availability

All other relevant data supporting the key findings of this study are available within the article and its Supplementary Information files. Source data are provided with this paper.

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

## Acknowledgements

We would like to publish this study in the loving memory of Dr. Barbara Murphy who contributed to the first part of this study. She passed away on June 30, 2021. J.C.H. is supported by NIH/NIDDK R01DK109683, R01DK122980, R01DK129467, P01DK56492, and VA Merit Award I01BX000345; K.L. is supported by NIH/NIDDK R01DK117913-01 and R01DK129467; C. Wei is supported by ASN Career Development Grant.

## Author contributions

C.W., J.C.H. and B.D. designed the research project. C.W., M.C., M.M., W.W., J.F., Z.Y., J.L., Z.L., L.M., K.B., S.L., Y.D., N.A., E.A. performed the experiments. B.D. designed and synthesized the compounds. C.W., J.C.H., W.Z., K.L., Z.Y., Z.S. analyzed the data. C.W., J.C.H., M.M., W.Z. and B.D. drafted and revised the manuscript. All authors approved the final version of the manuscript.

## Competing interests

The authors declare no competing interests.

## Additional information

[1]Division of Nephrology, Department of Medicine, Icahn School of Medicine at Mount Sinai, New York, NY, USA. [2]Department of Critical Care Medicine, Shandong Provincial Hospital affiliated to Shandong First Medical University, Jinan, China. [3]Department of Critical Care Medicine, Shandong Provincial Hospital, Shandong University, Jinan, China. [4]Division of Nephrology, Yale School of Medicine, New Haven, CT, USA. [5]Center for Comparative Medicine and Surgery, Icahn School of Medicine at Mount Sinai, New York, NY, USA. [6]Arnold and Marie Schwartz College of Pharmacy and Health Sciences, Long Island University, Brooklyn, NY, USA. [7]Renal Section, James J. Peters VAMC, Bronx, NY, USA. ✉e-mail: Bhaskar.Das@liu.edu; Cijiang.he@mssm.edu; Chengguo.wei@mssm.edu

