## [Peer Review File · Nature Communications]

HCK induces macrophage activation to promote renal inflammation and fibrosis via suppression of autophagyREVIEWER COMMENTS

Reviewer #1 (Remarks to the Author):

Chen et al describe the role for Hck in kidney inflammation. This report follows up previous work describing high levels of Hck kinase in renal biopsies from transplant rejection patients -- in mice inhibition of Hck with a broad spectrum kinase inhibitor (dasatinib) reduced kidney inflammation. They show that Hck is primarily expressed in macrophages, deletion of Hck affects many macrophage properties (from proliferation to migration to autophagy). Deletion of Hck (either globally or in myeloid cells) reduces kidney inflammation in two different models. The authors also describe the development of a new Hck kinase inhibitor, which also reduces inflammation in the kidney injury models.

This is an extensive study that represents a ton of work by the authors. There are some very novel aspects to the report -- the new Hck flx mouse (first one reported anywhere) and the new Hck kinase inhibitor. There is growing recognition of the involvement of Hck in a number of inflammatory disease processes. As such, I have only some minor suggestions that can help improve the manuscript.

1). The major weakness of this paper is the lack of mechanistic understanding of what signaling pathways Hck is involved in that lead to the cellular phenotypes described. Src family kinases transmit signals from extracellular receptors to intracellular signaling pathways. Think of Lck in the TCR pathway. Think of Src in EGFR signaling. Think of Hck and Fgr in integrin signaling. Think of Lyn in BCR and inhibitory (Sipr-alpha) signaling pathways. Nowhere in this paper do the authors contemplate what the signaling pathways Hck is involved in. Fig. 2 suggests that Hck is involved in LPS signaling leading to macrophage polarization. How does Hck (a tyrosine kinase) fit into TLR4 signaling (which is regulated by Ser/Thr kinases)? What are the upstream pathways involved in the differential autophagy responses described in Hck KO macrophages? The lack of thinking about pathways and mechanisms renders Fig. 2, 3, and 4 as mere descriptions of cell phenotypes without a mechanistic understanding. Unfortunately, this is common. One would think that the cell migration defect described in Hck knockout macrophages is due to the known role of this kinase in integrin pathways. Yet there are papers out there that describe normal migration of Hck-deficient myeloid cells. If the authors could spend some time in the Discussion thinking about receptors and signaling pathways, putting their findings into context with the literature, that would be helpful.

2). There are a number of papers describing contributions of Hck in cancer-associated inflammation. The authors should review these articles. This is especially true because one of them comes to the OPPOSITE conclusion as this report on the effect of Hck deletion on macrophage polarization (PMID: 36223746). The authors should contrast their results with others.

3). The figures and fig legends need work. Please use COLORS for all bar graphs in the manuscript. The tiny squares, circles are impossible to read. Stats are needed for Fig. 2B. Please explain Fig 4D; is it being compared to Fig. 2D? What are the blue lines in the scratch assays? What is Baf1 in Fig. 4A? Please define all the tissue/cell type abbreviations in Supp. Fig. 1. Consider expanding the figure legends to describe the experiments more completely.

4). Note that the LysM-cre used in this paper is NOT macrophage specific. It also deletes in neutrophils (PMID: 24857755). The authors need to change all the claims of macrophage-specific to myeloid cell specific. They should address this in the Discussion. All the in vivo affects of Hck inhibition or loss could be mediated as much by neutrophils as by macrophages.

Beyond these issues (most of which are minor) this is a strong paper, describing a new mouse with good in vivo data and a tantalizing birth announcement for a new compound. Hope it goes well.

Reviewer #2 (Remarks to the Author):

Renal inflammation and fibrosis are the common pathways leading to the progressive chronic kidney disease. Chen et al reported that hematopoietic cell kinase (HCK) was upregulated in human kidney fibrosis resulted from chronic allograft injury, and HCK expression is highly enriched in pro-inflammatory macrophages in diseased kidneys. HCK-knockout (KO) or HCK-inhibitor decreased macrophage M1-like pro-inflammatory polarization, proliferation, and migration in macrophages in vitro, and attenuated renal inflammation and fibrosis in UUO and IRI mice. They further developed a novel selective HCK inhibitor which reduces macrophage pro-inflammatory activity, proliferation, and migration in vitro, and attenuate kidney fibrosis in the UUO mice via autophagy related mechanisms. If the selective HCK inhibitor is safe and effective, it may be potentially significant as a new therapy for renal fibrosis.

Major comments:

1. The study showed that inhibition of HCK reduced renal fibrosis, however, it is unclear whether the renal function is improved in disease models. Serum creatinine and urinary protein levels should be measured in all study animals.
2. It is important to know the consequence of inhibition of HCK in normal animals, thus sham UUO/IRI mice treated with HCK inhibitor should be conducted.
3. Fig 1, I would suggest a HCK and CD68 double staining to show HCK expression in macrophages.
4. All interventions of HCK inhibition were conducted before the onset of renal injury, which are not the common scenario in clinical setting, these should be discussed in limitations.
5. Some grammar errors such as "stanning" (p16).

Point-to-Point response to the reviewers

We thank the reviewers for their constructive comments and suggestions, which significantly contribute to the improvement of our work. We have carefully addressed these questions and concerns in the revised version of the manuscript. In the following, the original comments and questions are echoed in black, and our response to each point is in blue. The underline highlighted data and text have been added to the revised manuscript. We also provide a revised manuscript with red color to highlight the changes.

Comments from the reviewers:

Reviewer #1

This is an extensive study that represents a ton of work by the authors. There are some very novel aspects to the report -- the new Hck flx mouse (first one reported anywhere) and the new Hck kinase inhibitor. There is growing recognition of the involvement of Hck in a number of inflammatory disease processes. As such, I have only some minor suggestions that can help improve the manuscript.

We thank the reviewer for the positive comments.

1). The major weakness of this paper is the lack of mechanistic understanding of what signaling pathways Hck is involved in that lead to the cellular phenotypes described.

Src family kinases transmit signals from extracellular receptors to intracellular signaling pathways. Think of Lck in the TCR pathway. Think of Src in EGFR signaling. Think of Hck and Fgr in integrin signaling. Think of Lyn in BCR and inhibitory (SIRP- α) signaling pathways. Nowhere in this paper do the authors contemplate what the signaling pathways Hck is involved in. Fig. 2 suggests that Hck is involved in LPS signaling leading to macrophage polarization. How does Hck (a tyrosine kinase) fit into TLR4 signaling (which is regulated by Ser/Thr kinases)? What are the upstream pathways involved in the differential autophagy responses described in Hck KO macrophages? The lack of thinking about pathways and mechanisms renders Fig. 2, 3, and 4 as mere descriptions of cell phenotypes without a mechanistic understanding. Unfortunately, this is common. One would think that the cell migration defect described in Hck knockout macrophages is due to the known role of this kinase in integrin pathways. Yet there are papers out there that describe normal migration of Hck-deficient myeloid cells. If the authors could spend some time in the Discussion thinking about receptors and signaling pathways, putting their findings into context with the literature, that would be helpful.

We thank the reviewer for pointing out this important issue. In addition to these known HCK-mediated signaling pathways, we thought that we identified a novel mechanism that HCK can directly interact with autophagy proteins ATG2A and CBL. Furthermore, in the revised manuscript, we performed additional experiments to study several known signaling pathways induced by HCK. We found that HCK KO in BMDM decreased the phosphorylation of PI3K/AKT signaling pathway, which is known to inhibit autophagy through mTOR. Therefore, we believe that HCK enhances autophagy activity via both regulation of PI3K/AKT pathway and direct interaction and phosphorylation of ATG2A and CBL. However, we did not see significant changes in the phosphorylation of Erk1/2 (p44/42 MAPK) and EGFR.

...Western blots (D) and quantification (E) showed phospho-PI3K and phospho-AKT decreased in HCK KO cells, compared to WT BMDMs. However, there were no significant differences for phospho-Erk1/2 and phospho-EGFR with HCK KO....

We also studied integrin signaling. We found that HCK KO inhibited BMDM cell spreading by F-actin staining. Tyrosine phosphorylation of SYK, a key kinase in integrin signaling, also decreased in HCK KO BMDMs. These data suggest that HCK regulates macrophages spreading and migration through integrin signaling, and suggests HCK has multiple roles in macrophage phenotype regulation. The reviewer mentioned “normal migration of Hck-deficient myeloid cells”. We think that the reviewer probably indicated the findings from the study published in Blood at 2010 (PMID: 19897576). This study showed that HCK doesn’t affect BMDMs 2D migration, but regulates BMDMs 3D migration, podosome organization and matrix degradation abilities. Our data suggests that HCK regulates both 2D and 3D migrations of BMDMs.

Figure S2. HCK KO in BMDMs impacted cell spreading by integrin signaling. Representative Immunofluorescence (IF) staining images (A) and quantification of cell spreading size (B) for F-actin in BMDM from WT and HCK KO mice. (C) Western blots for phospho-SYK show HCK KO decreased integrin signaling in BMDM.

As suggested, we also added a paragraph to discuss the signaling pathways of HCK in more details:

...we demonstrated a novel interaction between HCK and ATG2A and CBL, that may underlie its regulation of autophagy and show that HCK-KO increases autophagy activation in Raw264.7 and BMDMs. However, the mechanisms of autophagy regulation by HCK are not fully defined. Consistent with the previous findings, we identified that HCK KO in BMDMs decreased PI3K/AKT pathway (PMID: 15728479, 2005 J Immunol.; PMID: 27840303, 2016 Bio Biochim Biophys Acta), which is a known regulator of autophagy pathway through mTORC1. In addition, HCK likely plays multiple roles in cell signal transduction in macrophages. For instance, our data in BMDM and mouse models indicated HCK regulates macrophage migration. This is potentially through integrin signaling pathway, as we showed a decrease of F-actin spreading, phospho-SYK in HCK-KO cells. Independent studies have also shown that HCK regulates macrophage spreading and three-dimensional migration through integrin signaling pathway in NFS-60, BMDM and other myeloid cells (PMID: 9687507, 1998 EMBO Journal; PMID: 9197241, 1997 Current Biology; PMID: 10684859, 2000 J Exp Med; PMID: 19897576, 2010 Blood). Here, we also observed that HCK mediates LPS-induced macrophage M1 polarization, suggesting a role for HCK in regulating TLR signaling. Consistent with our data, other studies showed that HCK can regulate TLR signaling induced by TNF- \$\alpha\$ and IL-6 (PMID: 22021612, 2011 J Immunol.)

The EGF signaling pathway data are added to Figure 4, which describes HCK-regulated macrophage polarization through autophagy. The integrin signaling data are added to Supplementary Figure S2, showing that HCK regulates macrophage migration through integrin signaling.

2). There are a number of papers describing contributions of Hck in cancer-associated inflammation. The authors should review these articles. This is especially true because one of them comes to the OPPOSITE conclusion as this report on the effect of Hck deletion on macrophage polarization (PMID: 36223746). The authors should contrast their results with others.

We thank the reviewer for this important question. The role of the HCK in macrophage activity has been an area of much controversy, with conflicting findings in different cell types and disease conditions. It is likely that the role of HCK in cancer cells is different from non-cancer cells such as macrophages that we studied here. We have added the following discussion in the manuscript:

The overall role of the HCK in macrophage activity has been an area of much controversy, with conflicting findings in different cells and disease conditions. In the tumor associated macrophages (TAM), Poh et al showed that HCK's activity is associated with alternatively activated endotype(PMID: 28399411, 2017 Cancer Cell). Similar to this, others demonstrate that genetic ablation or chemical inhibition of HCK reprograms TAM to an inflammatory endotype and enhance T cell recruitment and activation (PMID: 36223746, 2022 Cell Reports; PMID: 35731867, 2022 Aci Adv; PMID: 31992566, 2020 Cancer Immunol Res.) However, HCK has opposite effects in non-cancer cells by promoting macrophage inflammatory activation. Ernst M, et al generated HCK constitutively active mutant mice and found that these mice developed spontaneous pulmonary inflammation and enhanced innate immune response (PMID: 12208875, 2002 J Exp Med). Later, they demonstrated that HCK, FGR, LYN are critical in generation of in vivo inflammatory environment in autoantibody-induced arthritis and other inflammation mouse models (PMID: 25225462, 2014 J Exp Med). Another study showed that loss of HCK/LYN increases M2 macrophage programming (PMID: 18246197, 2008 JCI). Our data are consistent with these latter findings, and show that inhibition or KO of HCK reduced pro-inflammatory polarization (M1-like) and increased M2-like polarization of macrophages, associating with reduced kidney injury and fibrosis in these animal models.

3). The figures and fig legends need work. Please use COLORS for all bar graphs in the manuscript. The tiny squares, circles are impossible to read. Stats are needed for Fig. 2B. Please explain Fig 4D; is it being compared to Fig. 2D? What are the blue lines in the scratch assays? What is Baf1 in Fig. 4A? Please define all the tissue/cell type abbreviations in Supp. Fig. 1. Consider expanding the figure legends to describe the experiments more completely.

We thank the reviewer for these suggestions, and we have changed all the figures and legends accordingly.

Due to the high cost of the cytokine antibody array, we only conducted a single experiment with one sample per group. However, the array panel included duplicate dots for each cytokine, allowing us to simultaneously measure multiple cytokines and assess the effects of HCK KO and inhibition on the overall profile of inflammatory cytokines. While we have also undertaken other studies to confirm the impact of HCK on macrophage inflammatory activity, this cytokine antibody array provides a comprehensive understanding of the cytokine profile affected by HCK.

Fig 4D shows HCK's effects on macrophage polarization, which is abrogated with treatment of PP242, an autophagy inducer. Figure 2D shows HCK KO decrease inflammatory activation (M1) and increase alternative activation (M2). These two figures indicate that HCK's effects on macrophage polarization are mainly regulated through autophagy. We have added this clarification to the revised manuscript.

The blue lines indicate the scratch edge of the cells when we measured the scratch distance generated by software to measure the scratch distance. We added this description to the legend.

Baf1 is bafilomycin A1. We explained all the abbreviations in the revised manuscript.

We provided all the abbreviations for the tissue/cell types in Supp. Fig.1 and for all other figures.

4). Note that the LysM-cre used in this paper is NOT macrophage specific. It also deletes in neutrophils (PMID: 24857755). The authors need to change all the claims of macrophage-specific to myeloid cell specific. They should address this in the Discussion. All the in vivo affects of Hck inhibition or loss could be mediated as much by neutrophils as by macrophages.

We thank the reviewer for pointing out this question. We changed macrophage-specific to myeloid cell specific in the revised manuscript and discussed this limitation in the manuscript as below:

LysM-cre is not macrophage specific because the LysM promoter also expresses in neutrophils (PMID: 24857755, 2014 J Immunol Methods). Previous studies suggest that HCK regulates neutrophil activation (PMID: 15723811, 2005 Immunity; PMID: 17339487, 2007 J Immunol) and migration (PMID: 9636192, 1998 PNAS; PMID: 31699792, 2020 Haematologica). Therefore, future studies are required to distinguish the roles of HCK in macrophages from neutrophils in kidney inflammation and fibrosis.

Beyond these issues (most of which are minor) this is a strong paper, describing a new mouse with good in vivo data and a tantalizing birth announcement for a new compound. Hope it goes well.

We thank the reviewer for this very positive feedback and providing very helpful comments.

Reviewer #2:

Major comments:

1. The study showed that inhibition of HCK reduced renal fibrosis, however, it is unclear whether the renal function is improved in disease models. Serum creatinine and urinary protein levels should be measured in all study animals.

We thank the reviewer for providing us with very helpful comments and suggestions. We measured the serum BUN and urinary protein levels in WT and HCK KO mice of uIRIx model. We found that HCK KO mice had improved kidney functions with decreased serum BUN and urinary protein levels. These data are now added in the revised Figure 6.

... (B) Urine ACR and serum BUN for WT and HCK KO uIRIx mice....

2. It is important to know the consequence of inhibition of HCK in normal animals, thus sham UUO/IRI mice treated with HCK inhibitor should be conducted.

We appreciate this important question from the reviewer. To test the toxicity of the HCK inhibitor, we daily gavaged WT mice with vehicle and BT424 for 1 month and did not observe any obvious toxicity based on the body weight changes, behavior, and physical activity. In addition, we performed Routine Chemistry Panel test in IDEXX BioAnalytics (Westbrook, ME 04092) with the serum from vehicle treated and BT424 treated mice and did not see noticeable changes in liver enzymes, renal function, lipid profile, and glucose levels. These data indicate that treatment of BT424 does not induce significant toxicity in these mice. We have added this information in the revised manuscript and this serum chemistry data to a new supplementary table.

Table S1. Serum chemistry test for mice treated with HCK selected inhibitor BT424 and vehicle

	PBS-1	PBS-2	PBS-3	PBS-4	BT424-1	BT424-2	BT424-3	BT424-4
ALP(U/L)	74	65	106	97	72	61	73	--
AST(U/L)	47	--	63	41	33	30	28	48
ALT(U/L)	15	22	28	27	23	16	14	15
Creatine kinase(U/L)	32	--	108	67	39	23	26	--
GGT(U/L)	0	--	0	0	0	0	0	0
Albumin(g/dL)	3.1	2.5	2.4	3.1	2.9	2.7	2.9	--
Total Bilirubin(mg/dL)	0.2	--	0.1	0.2	0.1	0.1	0.1	--
Total Protein(g/dL)	--	--	4.3	5.3	5.1	4.8	4.8	--
Globulin(g/dL)	--	--	1.9	2.2	2.2	2.1	1.9	--
Bilirubin-Conjugated(mg/dL)	0	--	0	0	0	0	0	--
BUN (mg/dL)	32	--	29	31	24	27	22	28
Creatinine(mg/dL)	0	--	0	0	0	0	0	--
Cholesterol(mg/dL)	95	--	53	81	78	78	67	--
Glucose(mg/dL)	109	--	192	230	223	160	236	--
Calcium(mg/dL)	9.3	--	7.5	9.2	9.1	8.4	8.5	--
Phosphorus(mg/dL)	7.3	--	7.2	8.7	7.3	5.9	8.3	--
Bicarbonate TCO2(mmol/L)	13	--	16	15	17	12	12	--
ALB/GLOB ratio	--	--	1.3	1.4	1.3	1.3	1.5	--
BUN/Creatinine Ratio	0	--	0.1	0.2	0.1	0	0	--
Bilirubin - Unconjugated(mg/dL)	0.2	--	0.1	0.2	0.1	0.1	0.1	--
Magnesium(mg/dL)	3.2	--	2.8	2.9	3.4	2.5	2.9	--
Triglycerides (mg/dL)	76	--	81	81	110	107	82	--
Hemolysis Index	Normal	Normal	Normal	Normal	Normal	Normal	Normal	Normal
Lipemia Index	Normal	Normal	Normal	Normal	Normal	Normal	Normal	Normal

HEMOLYSIS

- Normal, + Exhibits no significant effect on chemistry values.
- ++ Exhibits no significant effect on chemistry values.
- +++ May increase AST by 25-50% and decrease ALP and Direct Bilirubin by 25-50%.
- ++++ May increase AST and CPK by 25 -50%, decrease ALP by > 50%, decrease Total and Direct Bilirubin by 25-50%, and decrease SDMA by 10-25%.

LIPEMIA

- Normal, +, ++ Exhibits no significant effect on chemistry values.

+++ May decrease Direct Bilirubin by 25-50%.

++++ May decrease ALT, AST and Direct Bilirubin values by >50%.

3. Fig 1, I would suggest a HCK and CD68 double staining to show HCK expression in macrophages.

We thank the reviewer for this suggestion, and we performed co-staining of CD68 with pHCK or HCK in the biopsies. We found that CD68 and HCK co-localized in the macrophages. Figure 1B has been updated with this new staining.

...(B) IHC Staining of phosphorylated-HCK, HCK and CD68 in high chronic allograft damage index (CADI) and low CADI allograft kidneys...

4. All interventions of HCK inhibition were conducted before the onsite of renal injury, which are not the common scenario in clinical setting, these should be discussed in limitations.

We thank the review for pointing out this important question. We discussed this limitation as the reviewer suggested in the revised manuscript:

We used HCK inhibitor BT424 to treat the mice before the injury, indicating the prevention of kidney injury or fibrosis in these mouse models. Future studies are required to determine whether treatment of the mice after disease onset can reverse the kidney injury and fibrosis.

5. Some grammar errors such as ???stanning??? (p16).

We thank the reviewer for this suggestion, we went through the whole manuscript carefully and corrected all the grammar and spelling errors.

REVIEWER COMMENTS

Reviewer #1 (Remarks to the Author):

Chen, et al have submitted a revised version of their manuscript describing a role for the Hck kinase in renal inflammation/fibrosis. The authors have made a number of significant changes to the manuscript, including new experimental data, which address reviewer concerns. This has made a strong report even stronger. The changes are good and I have no further suggestions. The authors are to be congratulated on a fine paper.

Reviewer #2 (Remarks to the Author):

All my concerns have been addressed.

Reviewer #3 (Remarks to the Author):

The authors responded comprehensively to comments from reviewers 1 and 2 concerning biology part of the work. In my review, I will therefore focus on the part concerning medicinal chemistry and the development of a novel HCK inhibitor.

The importance of SAR studies of Dasatinib for the development of BT424 is not obvious taking into account that structure of BT424 containing the chromene scaffold is not related to the Dasatinib or its derivatives shown in Fig. 7A. As a result it does not contribute much to the understanding of the design of BT424 and the authors' idea in this regard.

To fill this gap please complete Fig. 7 with an analogous diagram as 7A showing examples referring to the structure of BT424 (BT294, 331, 332, 342 ?) and possibly the parent structure being a subject of optimization, or better, replace the current figure 7A with it. Please amend the section "Development of boron based pharmacophore group for HCK inhibitors with SAR studies of dasatinib" in the Results section accordingly.

In the Supplementary Material please provide the chemical structures of the synthesized 33 boron-based potential HCK inhibitors together with their code numbers.

Please change the sequential numbers of compounds in 7A to their code numbers, this will allow their identification in diagrams 7B-F.

The utility of boron in drug design is related mainly to the electronic and physicochemical properties of boron and its acids and their ability to mimic a tetrahedral sp³-hybridized carbon atom with appropriate substitution and under proper conditions (putting aside the use of boron clusters in drug design). The ability to reversibly form covalent bonds with alcohols adds another dimension. For this, two (Bortezomib, Ixazomib) or at least one (Vaborbactam, Crisaborole, Tvaborole) acidic hydroxyl group bonded to the boron atom are needed. However, BT424 is different. Please explain what is the rationale for using boron in the design of HCK inhibitors and what is the probable molecular base of BT424 activity. The crystal structure of HCK is known, and molecular docking may shed light on the superior properties of BT424.

Is BT424 sufficiently soluble in water? Please provide information on its stability in water-containing media.

Abstract. Boron is still not a common element used in drug design, please be more informative and change "Finally, we developed a novel HCK inhibitor..." to "Finally, we developed a novel, boron-containing HCK inhibitor..." or similar.

Page 10, lines 2 and 3, please change "phages" fort "phases".

In conclusion, I recommend publishing this important work after addressing the concerns above.

Point-to-Point response to the reviewers

We would like to thank the reviewers again for their comments and suggestions that helped us to improve our work. We have carefully addressed them in this response letter and the revised manuscript. In the following, the original comments and questions are echoed in black, and our response to each point is in blue. We also provide a revised manuscript with red color to highlight the changes.

Reviewer #1 (Remarks to the Author):

Chen, et al have submitted a revised version of their manuscript describing a role for the Hck kinase in renal inflammation/fibrosis. The authors have made a number of significant changes to the manuscript, including new experimental data, which address reviewer concerns. This has made a strong report even stronger. The changes are good and I have no further suggestions. The authors are to be congratulated on a fine paper.

Reviewer #2 (Remarks to the Author):

All my concerns have been addressed.

We thank these two reviewers for their constructive comments and suggestions, which significantly contribute to the improvement of our work.

Reviewer #3 (Remarks to the Author):

The authors responded comprehensively to comments from reviewers 1 and 2 concerning biology part of the work. In my review, I will therefore focus on the part concerning medicinal chemistry and the development of a novel HCK inhibitor.

The importance of SAR studies of Dasatinib for the development of BT424 is not obvious taking into account that structure of BT424 containing the chromene scaffold is not related to the Dasatinib or its derivatives shown in Fig. 7A. As a result it does not contribute much to the understanding of the design of BT424 and the authors' idea in this regard.

To fill this gap please complete Fig. 7 with an analogous diagram as 7A showing examples referring to the structure of BT424 (BT294, 331, 332, 342?) and possibly the parent structure being a subject of optimization, or better, replace the current figure 7A with it. Please amend the section ???Development of boron based pharmacophore group for HCK inhibitors with SAR studies of dasatininb??? in the Results section accordingly.

We thank the reviewer for this important question. We have replaced Fig 7A with an analogous diagram to show the structures of BT294, BT332, BT342 and BT424. We also show the parent structure for optimization of these compounds in the diagram. From Dasatinib we synthesized BT-294 lead compound (by changing thiazol and pyrimidinyl group of dasatinib in square to triazolo-thiadiazine derivative), then from BT294 we synthesized BT332(Thiazole amide derivatives), then we synthesized BT442 lead compound, then at end BT424 (oxadiazaborole derivatives).

New Fig 7A. SAR study of dasatinib to design novel HCK inhibitors.

In the Supplementary Material please provide the chemical structures of the synthesized 33 boron-based potential HCK inhibitors together with their code numbers.

We thank the reviewer for this question. We have provided all the 33 boron-based potential HCK inhibitors in supplementary Figure S4.

Please change the sequential numbers of compounds in 7A to their code numbers, this will allow their identification in diagrams 7B-F.

We thank the reviewer for pointing out this question. We added the sequential numbers of compounds in the new Fig 7A.

The utility of boron in drug design is related mainly to the electronic and physicochemical properties of boron and its acids and their ability to mimic a tetrahedral sp^3 -hybridized carbon atom with appropriate substitution and under proper conditions (putting aside the use of boron clusters in drug design). The ability to reversibly form covalent bonds with alcohols adds another dimension. For this, two (Bortezomib, Ixazomib) or at least one (Vaborbactam, Crisaborole, Tavorole) acidic hydroxyl group bonded to the boron atom are needed. However, BT424 is different. Please explain what is the rationale for using boron in the design of HCK inhibitors and what is the probable molecular base of BT424 activity. The crystal structure of HCK is known, and molecular docking may shed light on the superior properties of BT424.

We thank the reviewer for this critical question. Yes, we appreciate the reviewer's thought boron from sp^2 hybridization convert to sp^3 hybridization state as mimic of tetrahedral transition state as potential enzyme inhibitors. However heterocyclic compounds containing boron atom are an active area of research where oxadiazoles, and thiazole amides could be mimic to oxadiazaborole pharmacophore group. Completely new pharmacophore groups first-in-class drugs. Medicinal chemist Dr. Das published many papers regarding this and most recently this review article explained in detail:

Das BC, Adil Shareef M, Das S, Nandwana NK, Das Y, Saito M, Weiss LM. Boron-Containing heterocycles as promising pharmacological agents. *Bioorg Med Chem*. 2022 Jun 1;63:116748. doi: 10.1016/j.bmc.2022.116748. Epub 2022 Apr 18. PMID: 35453036.

As we focused on fragmented base drug discovery approach, so first focused on active pharmacophore groups then developed new pharmacophore groups around that.

Thanks to the reviewer for advising docking studies, we are in the process of doing this. As with all existing molecular software related to carbon atom docking, there is no suitable boron-based docking approach software available. Many times, boron-based docking gives false binding, but not accurate, so Dr. Das is developing his own software collaborating with Biovia to establish boron-based accurate docking. In the future, from our BT424, definitely we will use our approach to find new pharmacophore group. We are also in the process to get the crystal structure of HCK binding BT424.

Is BT424 sufficiently soluble in water? Please provide information on its stability in water-containing media.

We thank the reviewer for these important questions. BT424 is insoluble in water, we made the stock in DMSO with 100mM and diluted to 25 uM in medium to treat cells. Or made in 250 mg/ml in DMSO as stock, then was diluted to 2.5 mg/ml with PBS before gavage mice for final concentration of 25 mg/kg body weight. We performed stability test for BT424 by diluting 100mM stock in DMSO to 25 uM with water and keep it at room temperature for 24 hours, HPLC assay didn't find much degradation of BT424 (less than 3%).

We added this information to the updated manuscript with red color to highlight the changes.

Abstract. Boron is still not a common element used in drug design, please be more informative and change ???Finally, we developed a novel HCK inhibitor????? to ???Finally, we developed a novel, boron-containing HCK inhibitor????? or similar.

Page 10, lines 2 and 3, please change ???phages??? fort ???phases???

We thank the reviewer for these important questions. We have changed the MS with more informative words by "developed a novel, boron-containing HCK inhibitor" and changed the "phages" to "phases".

In conclusion, I recommend publishing this important work after addressing the concerns above.

We thank the reviewer for the suggestions and questions from the medical chemistry filed that significantly improved our manuscript.

REVIEWERS' COMMENTS

Reviewer #3 (Remarks to the Author):

All my concerns have been addressed.